# TACKLING NON-STATIONARITY IN REINFORCEMENT LEARNING VIA CAUSAL-ORIGIN REPRESENTATION

## ABSTRACT

In real-world scenarios, the application of reinforcement learning is significantly challenged by complex non-stationarity. Most existing methods attempt to model changes in the environment explicitly, often requiring impractical prior knowledge. In this paper, we propose a new perspective, positing that non-stationarity can propagate and accumulate through complex causal relationships during state transitions, thereby compounding its sophistication and affecting policy learning. We believe that this challenge can be more effectively addressed by tracing the causal origin of non-stationarity. To this end, we introduce the **C**ausal-**O**rigin **REP**resentation (**COREP**) algorithm. COREP primarily employs a guided updating mechanism to learn a stable graph representation for states termed as causal-origin representation. By leveraging this representation, the learned policy exhibits impressive resilience to non-stationarity. We supplement our approach with a theoretical analysis grounded in the causal interpretation for non-stationary reinforcement learning, advocating for the validity of the causal-origin representation. Experimental results further demonstrate the superior performance of COREP over existing methods in tackling non-stationarity.

## 1 INTRODUCTION

Rapid progress in reinforcement learning (RL) (Kaelbling et al., 1996; Sutton & Barto, 2018) in recent years has led to impressive performance improvements across various applications (Silver et al., 2018; Mirhoseini et al., 2020). However, a common assumption in many RL algorithms is the stationarity of the environment, which can limit their applicability in real-world scenarios characterized by varying dynamics (Padakandla et al., 2020; Padakandla, 2021). While recent meta-RL (Finn et al., 2017) methods have shown some promise in addressing non-stationarity through adaptation (Poiani et al., 2021), their performance often degrades when facing more complex changes in the dynamics (Sodhani et al., 2022; Feng et al., 2022). Several recent works, such as FN-VAE (Feng et al., 2022) and LILAC (Xie et al., 2020), have made strides towards improving RL algorithms in non-stationary environments by explicitly modeling the change factors of the environment. Nevertheless, they may not comprehensively capture the complexity of real-world non-stationarity.

In this paper, we propose a novel setting for efficiently tackling non-stationarity in RL from a new perspective inspired by the causality literature (Zhang et al., 2020; Huang et al., 2020). We argue that minor changes in dynamics can cause significant shifts in observations due to their propagation through intricate causal relationships among state elements. We believe that this challenge can be more effectively addressed by tracing the causal origin of non-stationarity. However, directly constructing an accurate causal graph of observations in non-stationary environments is challenging due to the instability of dynamics (Strobl, 2019). To overcome this challenge, we propose the **COREP** (**C**ausal-**O**rigin **REP**resentation) algorithm. COREP primarily utilizes a guided updating mechanism to learn a stable graph representation for states termed as causal-origin representation.

We first propose a novel formulation of non-stationarity in RL as a complex mixture of stationary environments. We take the masks defined in transition functions as causal relationships and assume that they are invariant within each environment. However, without access to any type of prior information about the environment, it is quite challenging to identify the causal relationship of each environment. Therefore, we consider an environment-shared representation defined by the union of maximal ancestral graphs (MAG) from each environment, and utilize it to design a stable RL

approach. In line with the concept of using the union of MAGs as the causal interpretation for non-stationary RL, we introduce a dual graph structure, termed core-graph and general-graph. The core-graph is designed for learning a stable graph representation, guided by a TD (Temporal Difference) error based updating mechanism. As the core-graph primarily concentrates on learning the most vital parts of the graph representation, some edges might be discarded in the process. Thus, a continuously updating general-graph is designed to compensate for this potential information loss and to improve the algorithm's adaptability. Ultimately, we integrate the core-graph and general-graph to construct the causal-origin representation, providing a comprehensive understanding of the environment's dynamics and significantly mitigating the impact of non-stationarity.

Specifically, our method starts by transforming observed states into node matrices. We then generate weighted adjacency matrices from the nodes to serve as inputs for two Graph Attention Networks (GAT) (Veličković et al., 2017), *i.e.*, core-GAT and general-GAT, representing the core-graph and general-graph, respectively. As a variant of Graph Neural Network (GNN) (Scarselli et al., 2008; Zhou et al., 2020), GAT can efficiently manage directed acyclic graphs (DAG) by prioritizing neighboring nodes via masked attention. This capability aligns perfectly with our goal of tracing the origins of non-stationarity through causal relationships. Then we determine whether to update the core-GAT by checking if the recent TD errors lie within the confidence interval of the replay buffer. The choice of using TD error as the metric for detection is based on the fact that significant changes in TD error imply notable alterations in the causal graph of observations, since TD error serves as a performance indicator for the learned policy in the current environment. This dual-GAT structure is then incorporated into a Variational AutoEncoder (VAE) (Kingma & Welling, 2013) to further enhance the learning efficiency. Our main contributions can be summarized as follows:

- We provide a causal interpretation for non-stationary RL and propose a novel setting that focuses on the causal relationships within states.

- Based on the proposed formulation and setting, we design a modular algorithm that can be readily integrated into existing RL algorithms.

- We provide a theoretical analysis that offers both inspiration and theoretical support for our algorithm. Experimental results further demonstrate the effectiveness of our algorithm.

## 2 PRELIMINARIES

**Problem Formulation.** Reinforcement learning problems are typically modeled as Markov Decision Processes (MDPs), defined as a tuple $(\mathcal{S}, \mathcal{A}, \mathcal{P}, \mathcal{R}, \gamma)$, where $\mathcal{S}$ is the state space, $\mathcal{A}$ is the action space, $\mathcal{P} : \mathcal{S} \times \mathcal{A} \times \mathcal{S} \rightarrow [0, 1]$ represents the transition probability, $\mathcal{R} : \mathcal{S} \times \mathcal{A} \rightarrow \mathbb{R}$ is the reward function, and $\gamma \in [0, 1)$ is the discount factor. We may also use the form of transition function $f : \mathcal{S} \times \mathcal{A} \rightarrow \mathcal{S}$ when the environment is deterministic. The goal of an agent in RL is to find a policy $\pi : \mathcal{S} \rightarrow \mathcal{A}$ that maximizes the expected cumulative discounted reward, defined as the value function $V^\pi(\boldsymbol{s}) = \mathbb{E}_\pi[\sum_{t=0}^\infty \gamma^t r_t | \boldsymbol{s}_0 = \boldsymbol{s}]$, where $r_t$ is the reward at time step $t$.

In non-stationary environments, the dynamics of the environment change over time. Our goal is to learn a policy $\pi$ that can adapt to the non-stationary environment and still achieve high performance. For simplicity of notations, we provide theoretical analysis with the form of deterministic transition function $f$, while the results still hold for the probabilistic setting.

**Causal Structure Discovery.** Causal structure discovery usually aims at inferring causation from data, modeled with a directed acyclic graph (DAG) $\mathcal{D} = (V, E)$, where the set of nodes $V$ includes the variables of interest, and the set of directed edges $E$ contains direct causal effects between these variables (Pearl et al., 2000). The causal graph is a practical tool that relates the conditional independence relations in the generating distribution to separation statements in the DAG (d-separation) through the Markov property (Lauritzen, 1996). If unobserved confounders exist, maximal ancestral graphs (MAGs) $\mathcal{M} = (V, D, B)$ are often used to represent observed variables by generalizing DAGs with bidirected edges which depicts the presence of latent confounders (Richardson & Spirtes, 2002). The sets $D, B$ stand for directed and bidirected edges, respectively.

**Graph Neural Networks (GNNs).** GNNs are a class of neural networks designed for graph-structured data. Given a graph $\mathcal{G} = (V, E)$, GNNs aim to learn a vector representation for each node $v \in V$ or the entire graph $\mathcal{G}$, leveraging the information of both graph structure and node

features. GNNs generally follow the message passing framework (Balcilar et al., 2021). Layers in GNN can be formulated as $\boldsymbol{H}^{(l)} = \sigma\left(\sum_s \boldsymbol{L}_s \boldsymbol{H}^{(l-1)} \boldsymbol{W}_s^{(l)}\right)$, where $\boldsymbol{H}^{(l)}$ is the node representation outputed by the $l$-th layer, $\boldsymbol{L}_s$ is the $s$-th convolution support which defines how the node features are propagated, $\boldsymbol{W}_s^{(l)}$ is learnable parameters for the $s$-th convolution support in the $l$-th layer, and $\sigma(\cdot)$ is the activation function. Graph Attention Network (GAT) (Veličković et al., 2017) is a special type of GNN following the message passing framework. Instead of handcrafting, the self-attention mechanism is used to compute the support convolutions in each GAT layer, where the adjacency matrix plays the role as a mask matrix for computing the attention.

## 3 METHODOLOGY

### 3.1 KEY IDEA

In our COREP algorithm, the primary goal is to address the issue of non-stationarity in RL by learning the underlying graph structure of the environment termed as causal-origin representation, which is desired to be causal and stable. This is achieved by designing a dual GAT structure, *i.e.*, a core-GAT and a general-GAT. The core-GAT is designed to learn the most essential parts of the environment's graph, and its learning is controlled by a guided updating mechanism. The general-GAT, on the other hand, is continuously updated to compensate for any information that the core-GAT might overlook. Together, they form a comprehensive understanding of the environment.

The causal-origin representation involves transforming states into node features and generating the weighted adjacency matrix. A self-attention mechanism is then applied to these nodes to compute the graph representation. To improve learning efficiency, the causal-origin representation is incorporated into the VAE framework. To guide the updating of the core-GAT, a TD error-based detection mechanism is employed. Furthermore, regularization terms are introduced to enhance the identifiability and guarantee the structure of the causal-origin representation.

### 3.2 CAUSAL INTERPRETATION FOR NON-STATIONARY RL

In this part, we will propose a causal interpretation for non-stationarity in RL, which provides us with inspiration and theoretical support for the algorithm design. We assume that the underlying dynamics can be described in terms of functions $f_i, g_j, k$:

$$
\begin{aligned}
s_i' &= f_i\left(\boldsymbol{c}_i^{s\to s} \odot \boldsymbol{s}, \boldsymbol{c}_i^{a\to s} \odot \boldsymbol{a}, \varepsilon_i^s\right), \forall i = 1, \cdots, d_s, \\
h_j' &= g_j\left(\boldsymbol{c}_j^{h\to h} \odot \boldsymbol{h}, \boldsymbol{c}_j^{s\to h} \odot \boldsymbol{s}, \boldsymbol{c}_j^{a\to h} \odot \boldsymbol{a}, \varepsilon_j^h\right), \forall j = 1, \ldots, d_h, \\
r &= k\left(\boldsymbol{c}^{s\to r} \odot \boldsymbol{s}, \boldsymbol{c}^{h\to r} \odot \boldsymbol{h}, \boldsymbol{c}^{a\to r} \odot \boldsymbol{a}, \varepsilon^r\right),
\end{aligned}
\tag{3.1}
$$

where $\boldsymbol{s}$ denotes the state with dimension $d_s$, $\boldsymbol{h}$ denotes the hidden state with dimension $d_h$, $r$ represents the reward, $s_i'$ is the $i$-th element of next state $\boldsymbol{s}'$, $h_j'$ is the $j$-th element of next hidden state $\boldsymbol{h}'$, and $\odot$ denotes the Hadamard product. Random noises $\varepsilon_i^s, \varepsilon_j^h, \varepsilon^r$ in Equation 3.1 are independent, and the distribution of each noise is identical. We consider the non-stationarity as the change of binary masks $\boldsymbol{c}^{\cdot\to\cdot}$. The masks represent the structural dependence in the following way. For example, in $f_i$, the $j$-the element of $\boldsymbol{c}_i^{s\to s} \in \{0,1\}^{d_s}$ equals 1 if and only if $s_j$ directly affects $s_i$. Other masks are defined in the same way. We rewrite the masks in matrix forms, letting $\boldsymbol{C}^{\cdot\to s} := [\boldsymbol{c}_i^{\cdot\to s}]_{i=1}^{d_s}$, $\boldsymbol{C}^{\cdot\to h} := [\boldsymbol{c}_j^{\cdot\to h}]_{j=1}^{d_h}$. The masks $\boldsymbol{C}^{\cdot\to\cdot}$ are allowed to be time-varying. Some previous work assumed that the masks are invariant across time, and encoded the changes over environments into some change factors (Huang et al., 2022). However, we do not rely on any assumptions of such change factors. Instead, we propose a novel causal interpretation based on an environment-shared union graph representation to capture the transition information in the non-stationary environment.

With the definition of the underlying dynamics, we can depict non-stationarity as being influenced by certain causal relationships. In other words, to represent these causalities, $\boldsymbol{s}$ is transformed into a new representation $\boldsymbol{h}$. Consequently, the original transition function $f_i$ is transformed into $g_j$, which is exactly the underlying dynamics that account for the causative factors of non-stationarity, as illustrated in Figure A.1. Under this setting, a non-stationary environment can be regarded as a non-stationary mixture of various stationary environments. Then the non-stationarity of the environment can be interpreted as the variations over time in this mixture distribution. We assume that there are

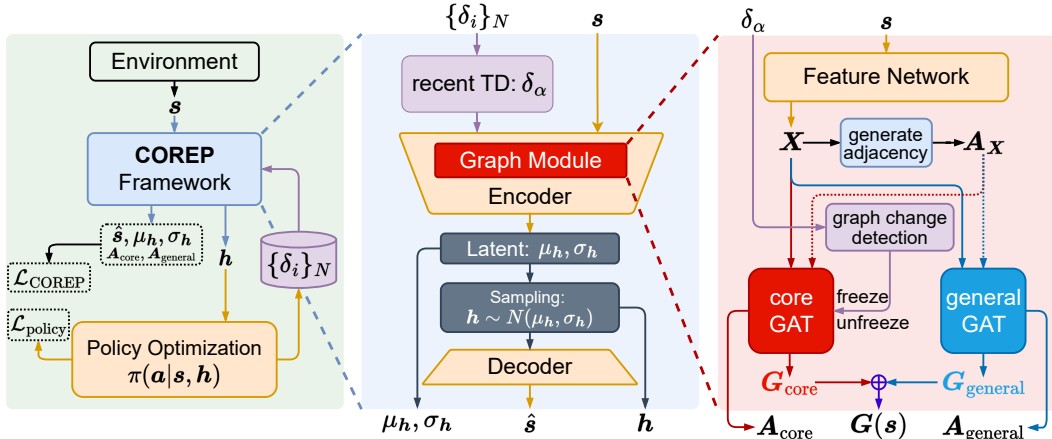

**Figure 3.1.** Overview of the COREP framework. (1) The left part illustrates that the COREP framework can be seamlessly incorporated into any RL algorithm. It takes the state as input and outputs the causal-origin representation for policy optimization. (2) The middle part shows the VAE structure employed by the COREP framework, which is utilized to enhance the learning efficiency. (3) The right part highlights the key components of COREP. The dual GAT structure is designed in line with the concept of causal-origin representation to retain the essential parts of the graph. The TD error detection can guide the core-GAT to learn the environment-shared union graph based on our theory. The general-GAT is continuously updated to compensate for the loss of information.

$K$ distinct stationary environments, where $K$ is an unknown number. If a sample is from the $k$-th environment, then there exists mask $\boldsymbol{C}_{(k)}^{\cdot\rightarrow\cdot}, k \in \{1, \ldots, K\}$, such that the current $\boldsymbol{C}_{\text{current}}^{\cdot\rightarrow\cdot} = \boldsymbol{C}_{(k)}^{\cdot\rightarrow\cdot}$, *i.e.*, the masks are invariant for each environment. The relationships among variables determined by masks can also be characterized by DAGs $\mathcal{D}_{(k)} = (V, E_{(k)})$ over the same set of nodes $V = \{\boldsymbol{s}, \boldsymbol{h}, \boldsymbol{a}, \boldsymbol{s}', \boldsymbol{h}', \boldsymbol{a}', r, e\}$, where $e \in \{1, \ldots, K\}$ is a node denoting the label of the environment with in-degree 0. The set of directed edges $E_{(k)}$ contains an edge from one node to another node if and only if the corresponding mask $\boldsymbol{c}_{(k)}^{\cdot\rightarrow\cdot} = 1$, and it also contains additional edges from $e$ to $(\boldsymbol{s}, \boldsymbol{h})$ whose marginal distributions vary across environments. In this setting, non-stationarity can be reflected by the changes in the underlying graph structure. To better understand the causal interpretation, we provide a specific toy example in Appendix C to illustrate such a setting.

## 3.3 UNION GRAPH OF THE CAUSAL STRUCTURE

In the context of causal interpretation, it is challenging to learn the structure of the causal graph independently when the environment label is unavailable, as the variable $e$ can be considered a latent confounder that leads to spurious correlations. In the presence of unobservable nodes, maximal ancestral graph (MAG) is a useful tool to generalize DAGs (Richardson & Spirtes, 2002). For each DAG $\mathcal{D}_{(k)}$, we can construct a corresponding MAG $\mathcal{M}_{(k)}$(Sadeghi, 2013), see Algorithm A.1. The MAGs make use of bidirected edges ($\leftrightarrow$) to characterize the change of marginal distribution of $\boldsymbol{s}, \boldsymbol{h}$ over different environments. To model structural relationships in non-stationary RL with a unified approach, we further encode the relations among all actions, states, hidden states, and rewards with an environment-shared union graph $\mathcal{M}_{\cup}$, which is defined as follows.

**Definition 3.1** (Environment-shared union graph). *The environment-shared representation union graph $\mathcal{M}_{\cup} := (V, D, B)$ has the set of nodes $V$, and the set of directed edges*

$$D = \{u \rightarrow v : u, v \in V, \exists k \text{ such that } u \rightarrow v \text{ in } \mathcal{M}_{(k)}\},$$

*and the set of bidirected edges*

$$B = \{u \leftrightarrow v : u, v \in V, \exists k \text{ such that } u \leftrightarrow v \text{ in } \mathcal{M}_{(k)}\}.$$

The above defined union graph $\mathcal{M}_{\cup}$ contains no cycle because Equation equation 3.1 implies that there exists a common topological ordering for $\mathcal{D}_{(1)}, \ldots, \mathcal{D}_{(K)}$, see Appendix A for details. Without

knowing the label of environment $k$, we cannot generally identify the structure of $\mathcal{M}_{(k)}$ for each $k$ from the observed data. However, we show that $\mathcal{M}_{\cup}$ is still a MAG, hence any non-adjacent pair of nodes is d-separated given some subset of nodes.

**Proposition 3.1.** *Suppose that the generative process follows Equation equation 3.1, then there exists a partial order $\pi$ on $V$ such that for all $i \in \{1, \ldots, K\}$, it holds that (a) $u$ is an ancestor of $v$ $\Rightarrow u <_\pi v$ in $\mathcal{M}_{(k)}$; and (b) $u \leftrightarrow v \Rightarrow u \not\lessgtr_\pi v$ in $\mathcal{M}_{(k)}$. As a consequence, the environment-shared representation union graph $\mathcal{M}_{\cup}$ is a MAG.*

We provide the full proofs and a detailed explanation of conditions in Appendix A. Proposition 3.1 provides theoretical support for recovering the structure of environment-shared union graph $\mathcal{M}_{\cup}$. In the following sections, we will describe how the COREP algorithm learns a policy that is stable under non-stationarity by utilizing the environment-shared representation union graph $\mathcal{M}_{\cup}$.

### 3.4 DUAL GRAPH ATTENTION NETWORK STRUCTURE

In accordance with Proposition 3.1, we aim to learn the causal-origin representation encapsulating the environment-shared union graph $\mathcal{M}_{\cup}$. For this purpose, we propose the structure of dual Graph Attention Networks (GAT), *i.e.*, core-GAT and general-GAT. The core-GAT is designed for learning the stable graph representation. We consequently control the update of core-GAT by employing TD error as a detector of significant changes in the environment's underlying graph structure. It is important to note that COREP uses the TD-error detection merely as an indicator of substantial environmental changes to decide whether or not to update the core-GAT, so we do not need a complex detection mechanism like other methods (Da Silva et al., 2006; Sutton et al., 2007) that are required to explicitly recognize the specific changes in the environment.

Since the core-GAT primarily focuses on learning the most essential part of the graph representation, some edges may be overlooked or lost in the process. To compensate for this potential loss of information and to enhance the algorithm's adaptation capabilities, we introduce the continuously updating general-GAT. In the end, we integrate the core-GAT and general-GAT to construct the causal-origin representation, thereby providing a comprehensive understanding of the environment's dynamics and significantly mitigating the impact of non-stationarity on decision-making in RL.

Specifically, we first transform states into node features using an MLP network $f_{\mathrm{MLP}} : \mathbb{R}^{d_s} \to \mathbb{R}^{N \cdot d_f}$, then reshape the output into node feature matrix $\boldsymbol{X} = \{\boldsymbol{x}_1, \boldsymbol{x}_2, \ldots, \boldsymbol{x}_N\} \in \mathbb{R}^{N \times d_f}$, where $N$ is the number of nodes, and $d_f$ is the number of features in each node $\boldsymbol{x}_i$. We then compute the weighted adjacency matrix which represents the probabilities of edges by using Softmax on the similarity matrix of nodes: $\boldsymbol{A_X} = \mathrm{Softmax}\left(\boldsymbol{X}\boldsymbol{X}^{\mathrm{T}} \odot (\boldsymbol{1}_N - \boldsymbol{I}_N)\right)$, where $\boldsymbol{1}_N \in \mathbb{R}^{N \times N}$ represents the matrix with all elements equal to 1, $\boldsymbol{I}_N \in \mathbb{R}^{N \times N}$ represents the identity matrix, and $\odot$ denotes the Hadamard product. Multiplying $(\boldsymbol{1}_N - \boldsymbol{I}_N)$ is to remove the self-loop similarity when computing the weighted adjacency matrix. A learnable weight matrix $\boldsymbol{W} \in \mathbb{R}^{d_f \times d_g}$ is then applied to the nodes for transforming $\boldsymbol{X}$ into graph features $\boldsymbol{X}\boldsymbol{W} \in \mathbb{R}^{N \times d_g}$ where $d_g$ denotes the dimension of the graph feature. We can then perform the self-attention mechanism on the nodes, *i.e.*, $\alpha_{ij} = \mathrm{attention}(\boldsymbol{x}_i\boldsymbol{W}, \boldsymbol{x}_j\boldsymbol{W}|\boldsymbol{A_X})$. The conditioned $\boldsymbol{A_X}$ allows us to perform the masked attention, *i.e.*, we only compute $\alpha_{ij}$ for node $j \in \mathbf{N}_i(\boldsymbol{A_X})$ where $\mathbf{N}_i(\boldsymbol{A_X})$ is the neighbor set of node $i$ computed by the weighted adjacency matrix $\boldsymbol{A_X}$. We can consider deeper-depth neighbors of each node by combining multiple $\mathrm{attention}(\cdot)$ into a multi-layer network. For the $n$-th graph attention layer, the coefficients computed by the self-attention mechanism can be specifically expressed as:

$$\alpha_{ij} = \frac{\delta_{\mathbf{N}_i(\boldsymbol{A_X})}(j) \cdot \exp\left(\sigma\left(\boldsymbol{l}_n\left[\boldsymbol{x}_i\boldsymbol{W} \oplus \boldsymbol{x}_j\boldsymbol{W}\right]^{\mathrm{T}}\right)\right)}{\sum_{k \in \mathbf{N}_i(\boldsymbol{A_X})} \exp\left(\sigma\left(\boldsymbol{l}_n\left[\boldsymbol{x}_i\boldsymbol{W} \oplus \boldsymbol{x}_k\boldsymbol{W}\right]^{\mathrm{T}}\right)\right)}, \tag{3.2}$$

where $\oplus$ is the concatenation operation, $\sigma$ is the activation function, $\boldsymbol{l}_n \in \mathbb{R}^{2d_g}$ is the learnable weight vector for the $n$-th graph attention layer, and $\delta_{\mathbf{N}_i(\boldsymbol{A_X})}(j)$ is the indicator function, *i.e.*, $\delta_{\mathbf{N}_i(\boldsymbol{A_X})}(j) = 1$ if $j \in \mathbf{N}_i(\boldsymbol{A_X})$ otherwise 0. We employ the multi-head attention (Vaswani et al., 2017) to stabilize the learning process. Specifically, we perform $M$ separate self-attention mechanisms as Equation (3.2) in each layer. Subsequently, the resulting features are concatenated to form the graph node:

$$\boldsymbol{g}_i = \overset{M}{\underset{m=1}{\oplus}} \sigma\left(\sum_{j \in \mathbf{N}_i(\boldsymbol{A_X})} \alpha_{ij}^{(m)} \boldsymbol{x}_j \boldsymbol{W}^{(m)}\right). \tag{3.3}$$

The core-GAT and general-GAT output $\boldsymbol{G}_{\text{core}}$ and $\boldsymbol{G}_{\text{general}}$, respectively, which are concatenated to form the final causal-origin representation. Specifically, we denote the entire process of obtaining the causal-origin representation from $\boldsymbol{s}$ as a function $\boldsymbol{G} : \mathbb{R}^{d_s} \to \mathbb{R}^{N \times 2Md_g}$, such that $\boldsymbol{G}(\boldsymbol{s}) \doteq \boldsymbol{G}_{\text{core}} \oplus \boldsymbol{G}_{\text{general}} = \{\boldsymbol{g}_1, \boldsymbol{g}_2, \ldots, \boldsymbol{g}_N\}_{\text{core}}^{\text{T}} \oplus \{\boldsymbol{g}_1, \boldsymbol{g}_2, \ldots, \boldsymbol{g}_N\}_{\text{general}}^{\text{T}}$.

We then feed the output $\boldsymbol{G}(\boldsymbol{s})$ into a Variational AutoEncoder (VAE) (Kingma & Welling, 2013) inference process to derive the latent representation $\boldsymbol{h}$. More details about this process can be found in Appendix B. It is noteworthy that the use of VAE serves solely as a tool to enhance learning efficiency, therefore it is not a strictly necessary component of our COREP algorithm. The latent $\boldsymbol{h}$ is then provided to the policy $\pi(\boldsymbol{a}|\boldsymbol{s}, \boldsymbol{h})$ for policy optimization. COREP does not restrict the choice of policy optimization algorithms. In our implementation, we choose the classic PPO algorithm (Schulman et al., 2017) for policy optimization.

## 3.5 GUIDED UPDATING FOR CORE-GAT

As the theoretical analysis in Section 3.2, our goal is to learn the causal-origin representation encapsulating the environment-shared union graph $\mathcal{M}_\cup$. To achieve this, we design a TD error-based detection mechanism to guide the updating of core-GAT. Specifically, we store the computed TD errors of policy optimization into the TD buffer $\mathcal{B}_\delta$. We compute the mean of recent TD errors as $\delta_\alpha = \left(\sum_{|\mathcal{B}_\delta| - \alpha|\mathcal{B}_\delta| < k < |\mathcal{B}_\delta|} \delta_k\right) / \alpha|\mathcal{B}_\delta|$, where $\alpha$ controls the proportion of recent TD errors for detection, and $|\mathcal{B}_\delta|$ indicates the number of elements in $\mathcal{B}_\delta$. We then check whether $\delta_\alpha$ lies within the confidence interval $(\mu_\delta - \eta\sigma_\delta, \mu_\delta + \eta\sigma_\delta)$, where $\mu_\delta, \sigma_\delta$ are the mean and standard deviation of the TD buffer, and $\eta$ represents the confidence level. If the recent TD error $\delta_\alpha$ lies within this interval, we freeze the weights of the core-GAT and halt its updates; otherwise, we unfreeze its weights and proceed with updating the core-GAT.

As we discussed previously, since the core-GAT primarily focuses on learning the most essential part of the graph representation, some edges might be discarded in the process. To compensate for the loss of information, we add a continuously updating general-GAT and guide the learning of core-GAT by introducing a regularization that penalizes the difference between the output adjacency matrices of the core-GAT and general-GAT:

$$\mathcal{L}_{\text{guide}} = \|\boldsymbol{A}_{\text{core}} - \boldsymbol{A}_{\text{general}}\|_2. \tag{3.4}$$

To enhance the identifiability, we introduce the regularization for the MAG structure and sparsity:

$$\mathcal{L}_{\text{MAG}} = \|\boldsymbol{A}_{\text{core}} - \boldsymbol{A}_{\text{core}}^{\text{T}}\|_2 + \|\boldsymbol{A}_{\text{general}} - \boldsymbol{A}_{\text{general}}^{\text{T}}\|_2,$$
$$\mathcal{L}_{\text{sparsity}} = \|\boldsymbol{A}_{\text{core}}\|_1 + \|\boldsymbol{A}_{\text{general}}\|_1, \tag{3.5}$$

where $\mathcal{L}_{\text{MAG}}$ is used for penalizing the asymmetry of the adjacency matrices to meet the design of the MAG which utilizes bidirected edges to characterize the change of marginal distribution of $\boldsymbol{s}, \boldsymbol{h}$ over different environments. Subsequently, we compute the total loss function by combining the objective of policy optimization $\mathcal{L}_{\text{policy}}$ with the aforementioned loss functions:

$$\mathcal{L}_{\text{total}} = \mathcal{L}_{\text{policy}} + \lambda_1 \mathcal{L}_{\text{guide}} + \lambda_2 (\mathcal{L}_{\text{MAG}} + \mathcal{L}_{\text{sparsity}} + \mathcal{L}_{\text{VAE}}), \tag{3.6}$$

where $\mathcal{L}_{\text{VAE}}$ is calculated according to Equation B.1. Considering that only $\mathcal{L}_{\text{guide}}$ has a significant difference in magnitude compared to other loss terms, we set a separate coefficient for it, while uniformly setting a common coefficient for the other loss terms. This ensures that COREP does not have difficulties in hyperparameter tuning. By computing and backpropagating the gradient of $\mathcal{L}_{\text{total}}$ to update the entire COREP framework, along with policy optimization in an end-to-end manner, we can prevent these distinct loss functions from resulting in abnormal causal structures. This ultimately ensures that the learned policy can effectively tackle non-stationarity. The overall framework is shown in Figure 3.1 and the detailed steps can be seen in Algorithm E.1.

## 4 EXPERIMENTS

In this section, we primarily aim to address the following questions: 1) Is COREP effective in addressing non-stationarity? 2) What is the contribution of each component in COREP? 3) Does COREP perform consistently under different degrees and settings of non-stationarity?

**Baselines.** To answer the above questions, we compare COREP with the following baselines: FN-VAE (Feng et al., 2022), VariBAD (Zintgraf et al., 2019), and PPO (Schulman et al., 2017). FN-VAE is the SOTA method for tackling non-stationarity, VariBAD is one of the SOTA algorithms in meta-RL that also has certain capabilities in handling non-stationarity, and PPO is a classical algorithm known for its strong stability. Furthermore, to examine the performance degradation caused by non-stationarity, we include an Oracle that has full access to non-stationarity information.

**Experimental settings.** We conducted experiments on various modified environments from the DeepMind Control Suite (Tassa et al., 2018). The DeepMind Control Suite is a widely-used benchmark for RL algorithms, and we modify it to introduce non-stationarity, enabling a comprehensive evaluation of our method. Due to the page limitation, we only present the performance results for 8 of them (*Cartpole Swingup, Reacher Hard, Cup Catch, Cheetah Run, Swimmer Swimmer6, Finger Spin, Fish Upright, and Quadruped Walk*) in our main manuscript. Full results and details of environments are provided in Appendix D. In our experimental settings, similar to FN-VAE, we introduce periodic noise to the dynamics to represent non-stationarity. To support our claim that COREP can handle more complex non-stationarity, we design a more intricate setting, *i.e.*, we randomly sample the coefficients for both within-episode and across-episode non-stationarity at every time step. Specifically, our modification can be expressed as:

$$s' = f(s, a) + f(s, a) \cdot \alpha_d \left[ c_1^t \cos(c_2^t \cdot t) + c_3^i \sin(c_4^i \cdot i) \right] \tag{4.1}$$

Here, $\alpha_d$ controls the overall degree of non-stationarity, and $c_k^t, c_k^i \sim \mathcal{N}(0.5, 0.5)$ represent the changing coefficients of within-episode and across-episode non-stationarity, respectively. This design generates various combinations of non-stationarity for each time step and episode, posing more significant challenges to our algorithm and the baselines.

**Performance.** As illustrated in Figure 4.1, the experimental results demonstrate that COREP outperforms all baselines in various environments, highlighting its consistent performance in the face of non-stationarity. Moreover, COREP exhibits significant performance improvements in more complex environments such as *Swimmer Swimmer6, Fish Upright, and Quadruped Walk*, accompanied by smaller variances, indicating its resilience against non-stationarity. Although VariBAD, as a meta RL method, can somewhat resist non-stationarity due to its ability of adaption, its large variances indicate insufficient stability. While the FN-VAE method, which directly models change factors, has shown performance similar to our COREP algorithm in certain simple environments, its performance in more complex environments proves that it cannot consistently handle complex non-stationarity. The narrower gap between COREP and Oracle further suggests that COREP effectively reduces the performance degradation caused by non-stationarity.

**Ablation study.** We conduct ablation studies to analyze the contribution of each component in COREP. To ensure consistency in our conclusion, the experiments are conducted under various non-stationarity settings, which include 'within-episode & across-episode', 'within-episode', and 'across-episode' non-stationarities. These settings are respectively denoted as (W+A)-EP, W-EP, and A-EP. As depicted in Figure 4.2a, eliminating the overall COREP structure and the TD error detection mechanism (corresponding to $\mathcal{L}_{\mathrm{guide}}$) from COREP both results in significant performance degradation, thereby substantiating the effectiveness of these two key designs in tackling non-stationarity. Moreover, the inclusion of VAE improves the learning efficiency. As expected, the regularization terms $\mathcal{L}_{\mathrm{MAG}}$ and $\mathcal{L}_{\mathrm{sparse}}$ also play important roles in enhancing the framework's ability to address non-stationarity.

Although removing these regularization terms can to some extent demonstrate the necessity of the highly coupled dual GAT structure in COREP, in order to further prove this point rigorously, we additionally investigate the performance when using only a single-GAT structure. Please refer to Appendix D.6 for specific results. Figure D.5 indicates that simply using a single GAT leads to significant performance degradation. In addition, we also investigate the sensitivity of COREP to the parameters in Equation 3.6. The detailed results are shown in Appendix D.7, which indicate that COREP is not very sensitive to these parameters, proving our claim that COREP does not require complex parameter tuning.

**Different degrees of non-stationarity.** As demonstrated by the consistent performance across different types of non-stationarities in Figure 4.2a, we also analyze the impact of varying degrees of non-stationarity in the environment, as depicted in Figure 4.2b. The results suggest that the performance of the compared baselines is more affected by the degree of non-stationarity. Conversely,

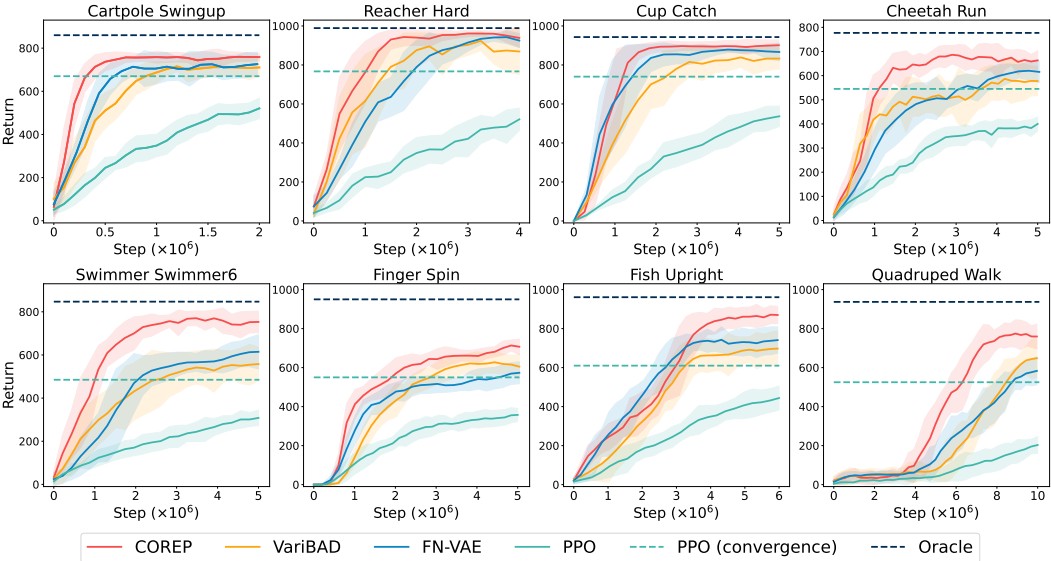

**Figure 4.1.** Learning curves of COREP and baselines in different environments. Solid curves indicate the mean of all trials with 5 different seeds. Shaded regions correspond to standard deviation among trials. The dashed lines represent the asymptotic performance of PPO and Oracle.

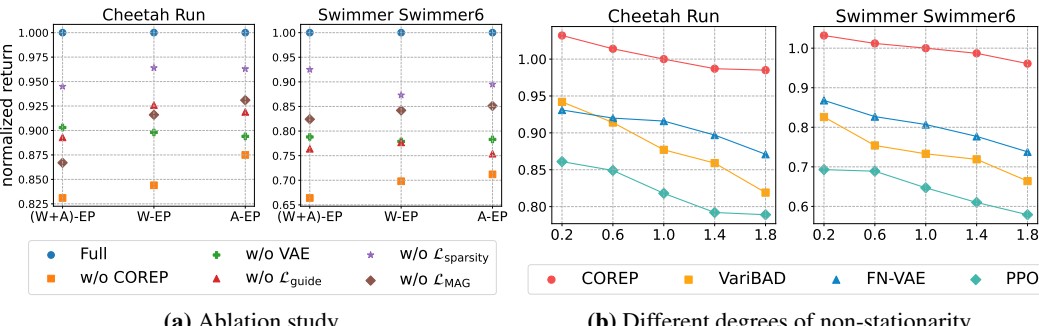

(a) Ablation study.

(b) Different degrees of non-stationarity.

**Figure 4.2.** Final mean returns of 3 different trials on Cheetah Run and Swimmer Swimmer6 environments with: (a) Different components and non-stationarity settings. Returns are normalized to the full version of COREP in each environment; (b) Different degrees of non-stationarity. Returns are normalized to the COREP algorithm with standard degree $1.0$.

COREP exhibits consistent performance when encountering different degrees of non-stationarity, further affirming our claim that COREP can effectively tackle more complex non-stationarity.

Due to page limitations, only partial experimental results are presented in the main manuscript. For full results, analyses, specific implementation details, and environmental settings, please refer to Appendix D and Appendix E.

## 5 RELATED WORK

**Non-stationary RL.** Pioneering research in non-stationary reinforcement learning primarily focused on detecting changes that had already occurred (Da Silva et al., 2006; Sutton et al., 2007), rather than anticipating them. Various methods have been developed to anticipate changes in non-stationary deep reinforcement learning settings. For example, Prognosticator (Chandak et al., 2020) maximized future rewards without explicitly modeling non-stationary environments, while MBCD (Alegre et al., 2021) employed change-point detection to determine whether an agent should learn a new policy or reuse existing ones. However, change-point detection may not work well in complex non-stationary environments and often requires providing priors. In cases where the evolution of non-stationary environments can be represented as a Semi-Markov chain, Hidden Markov-MDPs or Hierarchical Semi-Markov Decision Processes can be employed to address non-stationarity (Choi

et al., 1999; Hadoux et al., 2014). Some later work attempts to resist non-stationarity by leveraging the generalization of meta-learning (Finn et al., 2017). For example, Adaption via Meta-learning (Al-Shedivat et al., 2017) integrated continuous adaptation into the learning-to-learn framework to solve the non-stationarity. TRIO (Poiani et al., 2021) track non-stationarity by inferring the evolution of latent parameters, capturing the temporal change factors during the meta-testing phase. GrBAL (Nagabandi et al., 2018) meta-trained dynamic priors, enabling efficient adaptation to local contexts. However, these methods require the pre-definition of non-stationary tasks and subsequent meta-training on them. In real-world scenarios, though, we cannot access information about the non-stationarity. An alternative line of research directly learns latent representations to capture non-stationary components, leveraging latent variable models to directly model change factors in environments or estimating latent vectors describing the non-stationary or variable aspects of dynamics. LILAC (Xie et al., 2020) regarded the change factor as a latent variable and explicitly modeled the latent MDP. FN-VAE (Feng et al., 2022) modeled multiple latent variables of non-stationarities to achieve better performance. However, in real-world scenarios, non-stationarity itself is often more complex. Simply modeling the latent dynamics may not solve such complex scenarios well. Interpreting non-stationarity from a causal perspective is another novel approach. In addition, some work learns controllers on a collection of pre-defined stationary environments (Provan et al., 2022; Deng et al., 2022; Zhang & Li, 2019), which can get a guaranteed controller for any mixture of these stationary environments, thereby improving performance in more complex environments. Although our method theoretically treats non-stationary environments as mixtures of discrete stationary environments in a similar way, we use a more elegant update mechanism in practical design to ensure that our method is applicable to continuously changing general non-stationary environments. Some other work like (Saengkyongam et al., 2023) attempts to find an invariant causal structure to address non-stationarity, which shares some similar points with us. However, their method relies on offline data and has only been verified in simple Contextual Bandits environments. In contrast, our COREP algorithm can do online learning and tackle non-stationarity in complex environments.

**Causal structure learning.** Various approaches for learning causal structure from observed data have been proposed, see (Vowels et al., 2022) for a review. These approaches mainly fall into two broad categories: constraint-based methods and score-based methods. The constraint-based methods check the existence of edges by performing conditional independence tests between each pair of variables, *e.g.* PC (Spirtes et al., 2000), IC (Pearl et al., 2000), and FCI (Spirtes et al., 1995; Zhang, 2008). In contrast, the score-based methods generally view causal structure learning as a combinatorial optimization problem, and measure the goodness of fit of graphs over the data with a score, then optimize such score to find an optimal graph or equivalent classes (Chickering, 2002; Koivisto & Sood, 2004; Silander & Myllymäki, 2006; Cussens et al., 2017; Huang et al., 2018). Recently, some gradient-based methods which transform the discrete search into a continuous optimization by relaxing the space over DAGs have been proposed. These methods allow for applying continuous optimizations such as gradient descent to causal structure learning. For example, NOTEARS (Zheng et al., 2018) reformulated the structure learning problem as a continuous optimization problem, and ensured acyclicity with a weighted adjacency matrix. Dag-gnn (Yu et al., 2019) proposed a generative model parameterized by a GNN and apply a variant of the structural constraint to learn the DAG. Some researchers (Saeed et al., 2020) considered the distribution arising from a mixture of causal DAGs, used MAGs to represent DAGs with unobserved nodes, and showed the identifiability of the union of component MAGs. In this work, we characterize the non-stationarity by the union of MAGs, which enlightens us to design the COREP algorithm modelled by GNN.

## 6 Conclusions, Limitations and Future Work

In this work, we present a causal interpretation for non-stationary RL and propose a novel algorithm that focuses on the causal relationships within states. This algorithm learns a causal-origin representation to tackle more complex non-stationarity problems. Grounded in our proposed formulation, we design a modular algorithm that can be seamlessly integrated with existing RL algorithms. The theoretical analysis provides both inspiration and theoretical support for our algorithm. Experimental results from various non-stationary environments further demonstrate the efficacy of our algorithm.

However, our method does have certain limitations. The current approach may encounter scalability issues in high-dimensional state spaces, as the graph-based representation could be computationally intensive. In future work, we aim to address this issue by incorporating causal-origin representation learning into other types of latent variable models, including normalizing flows and probabilistic graphical models. This could further enhance both the scalability and performance of our method.

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

## A   CAUSALITY BACKGROUND AND PROOFS

We first review the definition of the Markov condition, the faithfulness assumption and some graphical concepts shown in the condition of Theorem 3.1. We use $\mathrm{pa}_{\mathcal{D}}(v)$, $\mathrm{ch}_{\mathcal{D}}(v)$, and $\mathrm{an}_{\mathcal{D}}(v)$ to denote the parents, children and ancestor of node $v$, respectively; for the detailed definitions, see *e.g.* (Lauritzen, 1996).

**Definition A.1** (Global Markov Condition (Pearl et al., 2000))**.** *A distribution $P$ over $V$ satisfies the global Markov condition on graph $\mathcal{D}$ if for any partition $(X, Y, Z)$ such that $X$ is d-separated from $Y$ given $Z$, then $X$ and $Y$ are conditionally independent given $Z$.*

**Definition A.2** (Faithfulness (Pearl et al., 2000))**.** *There are no independencies between variables that are not entailed by the Markov Condition.*

Under the above assumptions, we can tell the conditional independences using the d-separation criterion from a given DAG $\mathcal{D}$ (Pearl et al., 2000). Similarly, the MAGs are ancestral graphs where any non-adjacent pair of nodes is d-separated (Richardson & Spirtes, 2002). The following algorithm shows how to construct a MAG from DAG (Saeed et al., 2020):

---

**Algorithm A.1** Construction of the maximal ancestral graph

---

 1: Input: DAG $\mathcal{D} = (V, E)$
 2: Initialize $D = \emptyset, B = \emptyset$
 3: **for** $u, v \in \mathrm{ch}_{\mathcal{D}}(y)$ **do**
 4:     add $u \leftrightarrow v$ to $B$.
 5: **end for**
 6: **for** $t, u, v$ such that $(t \rightarrow u) \in E$ and $(u \leftrightarrow v) \in B$ **do**
 7:     **if** $u \in \mathrm{an}_{\mathcal{D}}(v)$ **then**
 8:         add $t \rightarrow v$ to $D$
 9:     **end if**
10: **end for**
11: **for** $u, v$ such that $(u \leftrightarrow v) \in B$ **do**
12:     **if** $u \in \mathrm{an}_{\mathcal{D}}(v)$ **then**
13:         remove $u \leftrightarrow v$ from $B$ and add $u \rightarrow v$ to $D$
14:     **end if**
15: **end for**

---

To illustrate the above algorithm, we provide two figures. Figure A.1 shows the underlying causal DAGs for the two environments, and Figure A.2 depicts the output of Algorithm A.1 as well as the corresponding environment-shared union graph.

In our paper, we characterize the nonstationarity as a mixture of stationary distributions. Formally, we take the following definition.

**Definition A.3** (Mixture of stationary distributions)**.** *The marginal distribution of $\{s, h, a, s', h', a'\}$ is a mixture of stationary distributions across environments, i.e.,*

$$P(s, h, a, s', h', a') = \sum_{k=1}^{K} \pi_k P(s, h, a, s', h', a' \mid e = k),$$

*where $\pi_k$ denotes the probability that the sample is from the $k$-th environment varying over time, and $P(s, h, a, s', h', a' \mid e = k)$ is invariant over time.*

*Proof of Proposition 3.1.* The outline of the proof are as follows. We first construct a strict partial order $\pi$ on $V$. Then, we induce the MAGs $\mathcal{M}_{(1)}, \ldots, \mathcal{M}_{(k)}$ from the DAGs $\mathcal{D}_{(1)}, \ldots, \mathcal{D}_{(k)}$ by applying the rules defined in Algorithm A.1. We show the constructed partial order $\pi$ is *compatible*, that for all $1 \leq k \leq K$, it holds that (a) $u \in \mathrm{an}(v) \Rightarrow u <_{\pi} v$ in $\mathcal{M}^{(k)}$; and (b) $u \leftrightarrow v \Rightarrow u \not\lessgtr_{\pi} v$ in $\mathcal{M}^{(k)}$. Finally we leverage the existing results in (Saeed et al., 2020) to conclude that $\mathcal{M}_{\cup}$ is a MAG.

We define a relation $\pi$ on $V$ as following: for any variable $u \in \{s, h, a\}$ and any variable $v \in \{s', h', a'\}$, we have (i) $u <_{\pi} v$; (ii) $v <_{\pi} r'$. To show the above defined $\pi$ is a strict partial order,

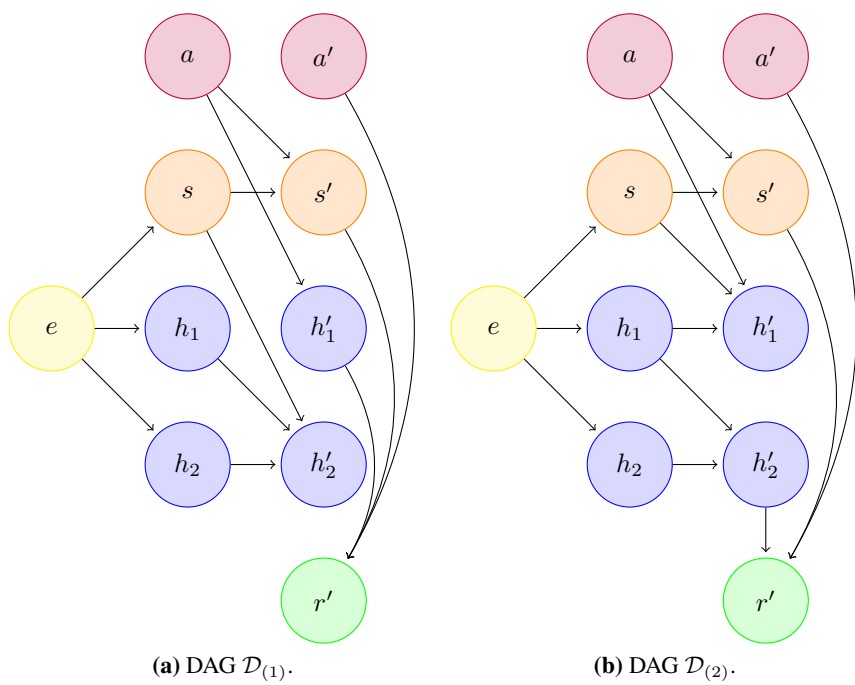

**(a)** DAG $\mathcal{D}_{(1)}$.      **(b)** DAG $\mathcal{D}_{(2)}$.

**Figure A.1.** DAG representations for two different environments.

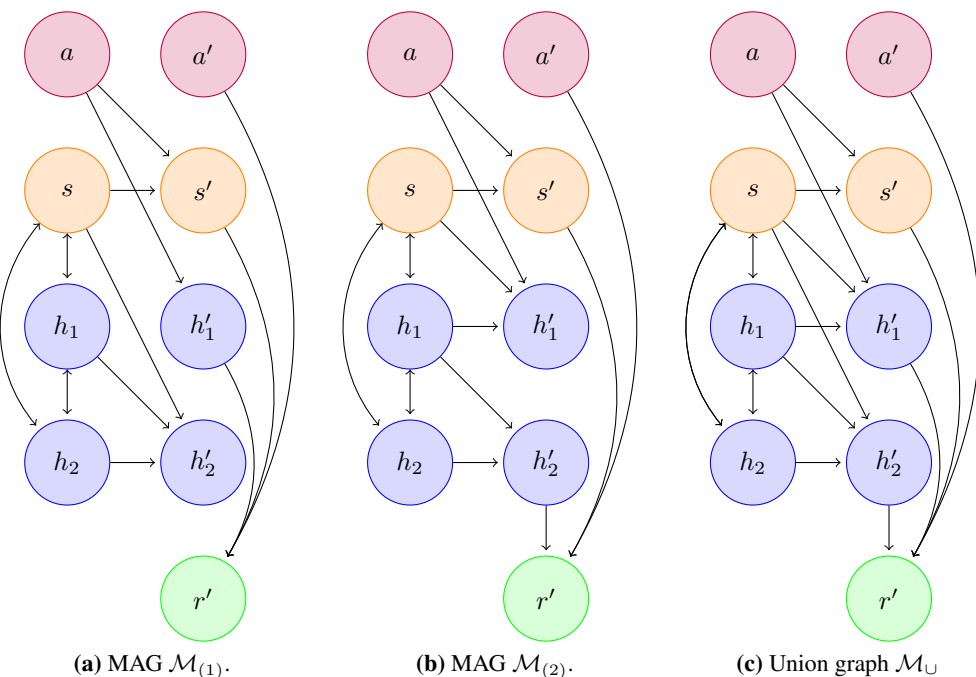

**(a)** MAG $\mathcal{M}_{(1)}$.      **(b)** MAG $\mathcal{M}_{(2)}$.      **(c)** Union graph $\mathcal{M}_{\cup}$

**Figure A.2.** MAG representations for two different environments and the union graph of two environments.

we first notice that $\pi$ is irreflexive, because $u \nprec_\pi u, v \nprec_\pi v$ and $r' \nprec_\pi r'$. The transitivity and asymmetry also hold by definition of $\pi$. Therefore, $\pi$ is a partial order on $V$.

The Algorithm A.1 constructs an MAG from DAG with three steps. The first step is to add bidirected edges among the nodes in $\text{ch}(e)$. Different values of $e$ leads different marginal distribution of $\boldsymbol{s}, \boldsymbol{h}$, hence $\text{ch}(y) \subseteq \{\boldsymbol{s}, \boldsymbol{h}\}$. Therefore, the bidirected edges are added with both nodes belonging to $\{\boldsymbol{s}, \boldsymbol{h}\}$. For the second step, there is no such node $t$, with $(t \to u) \in E$ and $(u \leftrightarrow v) \in B$, because the nodes in $\{\boldsymbol{s}, \boldsymbol{h}\}$ have no ancestor other than itself. So the second step adds the directed edges when $u = v$. The third step in our case is redundant. Equation equation 3.1 shows that there is no instantaneous causal effects in the system, so there is no $u, v$ such that $(u \leftrightarrow v) \in B$ while $u \in \text{an}_\mathcal{D}(v)$. From all above, if the input of Algorithm A.1 is $\mathcal{D}_{(k)}$, then it outputs a MAG $\mathcal{M}_{(k)} = (V, D_{(k)}, B_{(k)})$ with the set of nodes $V$, the set directed edges $D$ equals to the set of directed edges $E_{(k)}$ after removing the node $e$, and the set of bidirected edges $B_{(k)}$ consists edges among nodes in $\{\boldsymbol{s}, \boldsymbol{h}\}$.

Then, we check the condition (a) and (b) to show $\mathcal{M}_{(1)}, \ldots, \mathcal{M}_{(k)}$ are compatible with the above defined $\pi$. For (a), if $u$ is the ancestor node of $v$, then the structure of $\mathcal{M}_{(k)}$ implies that either $u \in \{\boldsymbol{s}, \boldsymbol{h}\}$ and $v \in \{\boldsymbol{s}', \boldsymbol{h}'\}$ are nodes in $\{\boldsymbol{s}', \boldsymbol{h}'\}$, or $u$ is a node from $\{\boldsymbol{s}', \boldsymbol{h}'\}$ and $v = r'$. For (b), if $u \leftrightarrow v$, then $u, v$ are nodes in $\{\boldsymbol{s}, \boldsymbol{h}\}$, hence $u \nlesseqgtr_\pi v$ in $\mathcal{M}^{(k)}$. These means that $\pi$ is a common strict partial order on $V$ for all MAGs. In this setup, we can leverage existing results from Lemma 4.3 (Saeed et al., 2020) to show that the environment-shared union graph $\mathcal{M}_\cup$ is also a maximal ancestral graph.

$\square$

# B  INCORPORATION WITH VARIATIONAL AUTOENCODER

To improve the efficiency of learning the causal-origin representation, we incorporate the causal-origin representation into the Variational AutoEncoder (VAE) framework (Kingma & Welling, 2013). Specifically, we feed the output $\boldsymbol{G}(\boldsymbol{s})$ into the VAE inference process to derive the mean and variance $(\mu_{\boldsymbol{h}}, \sigma_{\boldsymbol{h}})$ of the latent representation $\boldsymbol{h}$. Subsequently, we can sample $\boldsymbol{h} \sim \mathcal{N}(\mu_{\boldsymbol{h}}, \sigma_{\boldsymbol{h}}), \boldsymbol{h} \in \mathbb{R}^{d_h}$. The loss function for VAE is defined as

$$
\begin{aligned}
\mathcal{L}_{\text{VAE}}(\boldsymbol{s}; \theta, \phi) &= \mathbb{E}_{q_\phi(\boldsymbol{h}|\boldsymbol{G}(\boldsymbol{s}))} \left[\log p_\theta(\boldsymbol{G}(\boldsymbol{s})|\boldsymbol{h})\right] - \text{KL}\left[q_\phi(\boldsymbol{h}|\boldsymbol{G}(\boldsymbol{s}))||p(\boldsymbol{h})\right] \\
&\approx \text{MSE}(\boldsymbol{s}, \hat{\boldsymbol{s}}) - \text{KL}\left[q_\phi(\boldsymbol{h}|\boldsymbol{G}(\boldsymbol{s}))||\mathcal{N}(0, \boldsymbol{I})\right],
\end{aligned}
\tag{B.1}
$$

where $p_\theta, q_\phi$ represent the parameterized decoder and encoder respectively, $\text{KL}(\cdot)$ denotes the Kullback-Leibler divergence, and $\text{MSE}(\boldsymbol{s}, \hat{\boldsymbol{s}})$ is an estimation of $\mathbb{E}_{q_\phi(\boldsymbol{h}|\boldsymbol{G}(\boldsymbol{s}))}\left[\log p_\theta(\boldsymbol{G}(\boldsymbol{s})|\boldsymbol{h})\right]$ which measures the mean square error between the original state and the reconstructed state with the causal-origin representation. It is noteworthy that the VAE structure serves solely as a tool to enhance learning efficiency, therefore it is not a strictly necessary component of our COREP algorithm. The latent $\boldsymbol{h}$ is then provided to the policy $\pi(\boldsymbol{a}|\boldsymbol{s}, \boldsymbol{h})$ for policy optimization.

## C  Toy Example under the Causal Interpretation

To better understand the causal interpretation for non-stationary RL, let's consider a simple toy example. Given a stationary environment with a state space represented as $(s_1, s_2)$. In this example, to maintain simplicity, we focus only on the state's mask, omitting the action mask and noise term. We define the original dynamics function as $f(\boldsymbol{c}_s \odot \boldsymbol{s}, a) = \boldsymbol{c}_s \odot \boldsymbol{s} + a$. For the toy environment, we consider a basic causal model wherein $s_i'$ is only influenced by $s_i$.

Consequently, the original mask is

$$\boldsymbol{c}_s = \begin{pmatrix} 1 & 0 \\ 0 & 1 \end{pmatrix} \tag{C.1}$$

Given this, we can derive that

$$\begin{aligned} s_1' &= s_1 + a \\ s_2' &= s_2 + a. \end{aligned} \tag{C.2}$$

We denote the non-stationarity in our experiments (Eq 4.1) simplistically as $\boldsymbol{s}' = f(\boldsymbol{s}, a)[1 + n(t)]$, where $n(t)$ represents the introduced non-stationarity, which makes the dynamics becoming time-varying. In this scenario, the non-stationary environment's dynamics function becomes

$$\begin{aligned} s_1' &= s_1 + a + (s_1 + a)n(t) \\ s_2' &= s_2 + a + (s_2 + a)n(t). \end{aligned} \tag{C.3}$$

It is obvious that the dynamics introduces a time-varying term. In this context, we can define $h_i \doteq (s_i + a)n(t)$, leading to $s_i' = s_i + h_i + a$. This allows us to deduce the dynamics of $\boldsymbol{h}$ as

$$\begin{aligned} h_i' &= (s_i' + a) \cdot n(t+1) \\ &= (s_i + h_i + 2a) \cdot n(t+1) \\ &\doteq g_i(\boldsymbol{c}_s^t \odot \boldsymbol{s}, \boldsymbol{c}_h^t \odot \boldsymbol{h}, a), \end{aligned} \tag{C.4}$$

where $\boldsymbol{c}_s^t, \boldsymbol{c}_h^t$ symbolize the time-varying masks caused by non-stationarity $n(t)$.

More specifically, we derive

$$\boldsymbol{c}_s^t = \boldsymbol{c}_h^t = \begin{pmatrix} n(t+1) & 0 \\ 0 & n(t+1) \end{pmatrix} \tag{C.5}$$

The masks represent the causal effects between variables after non-stationary changes occur in this example. The values represent the degree of causal effects, which can be treated as edge weights after normalization. Based on our theory, COREP's goal is to learn the union graph shared by these graph structures and edge weights during the change process. That is, through a common graph with edge weights, it includes all possible non-stationary changes. Therefore, the information of non-stationarity in the final learned graph is reflected both in the topology of the union graph and in the edge weights representing probabilities.

By introducing the time-varying masks and $\boldsymbol{h}$, we can make the dynamics function remains stationary, transferring non-stationarity to the causal model. Thus, we have provided a walk-through under the simple toy example.

In fact, as illustrated in Figures A.1 and A.2, there are more intricate causal relationships in complex environments. As depicted above, $\boldsymbol{h}$ can encapsulate not only the inherent environmental information but also the complex causal relationships with non-stationarity. Our proposed union MAG (Proposition 3.1) and the correspondingly designed dual-GAT architecture aim to learn such intricate causal models, enabling generic RL algorithms to handle non-stationarity under this causal representation.

# D FULL EXPERIMENT DETAILS

## D.1 DETAILS ABOUT ENVIRONMENT SETTINGS.

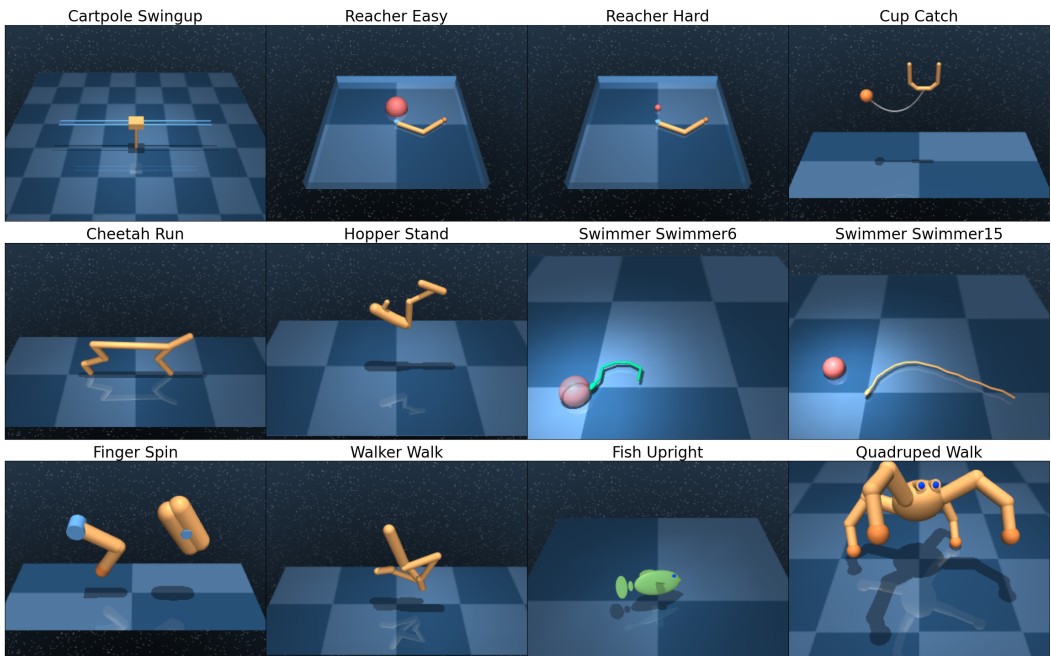

**Figure D.1.** The environment we use in our experiment. We add non-stationary noise to the observations of these environments according to Equation 4.1.

Figure D.1 shows the environments we use in the experiment. We add non-stationary noise to the observations of these environments according to Equation 4.1. These environments vary in terms of complexity, from low-dimensional problems like "Reacher Easy" to high-dimensional ones like "Quadruped Walk". All these tasks require the agent to understand and control its physical embodiment in order to achieve the desired goals. The specific descriptions of these environments and goals are as follows.

**Cartpole Swingup.** The cart can move along a one-dimensional track. The pole is attached to the cart with a joint allowing it to rotate freely. The initial state has the pole hanging down, and the goal is to apply forces to the cart such that the pole swings up and is balanced upright. Actions typically involve applying a horizontal force to the cart.

**Reacher Easy.** The agent is a two-joint robotic arm. The arm must move in a two-dimensional plane to touch a target position. The arm's state includes its joint angles and velocities. The action is the torque applied to each of the joints. The target's position is fixed in this version.

**Reacher Hard.** The task is the same as "Reacher Easy," but the target position is randomly placed in each episode, making the task more difficult as the agent has to learn to reach various positions.

**Cup Catch.** The agent is a robotic arm holding a cup, and there's a ball attached to the cup with a string. The arm needs to move in a way to swing the ball and catch it in the cup. The arm's state includes the position and velocity of the arm joints and the position and velocity of the ball. The actions are the torques applied at the arm's joints.

**Cheetah Run.** The agent is a model of a cheetah-like robot with 9 DoF(Degrees of Freedom): the agent can flex and extend its "spine," and each leg has two joints for flexing and extending. The agent's state includes the joint angles and velocities, and the actions are the torques applied to each of the joints. The goal is to move forward as fast as possible.

**Hopper Stand.** The agent is a one-legged robot, and its goal is to balance upright from a resting position. The agent's state includes the angle and angular velocity of the torso, as well as the joint angles and velocities. The actions are the torques applied to the joints.

**Swimmer Swimmer6.** The agent is a snake-like robot swimming in a two-dimensional plane. The robot has 6 joints, and the goal is to swim forward as fast as possible. The agent's state includes the joint angles and velocities, and the actions are the torques applied to the joints.

**Swimmer Swimmer15.** This is a more complex version of the Swimmer environment, with the agent being a 15-joint snake-like robot. Like the simpler version, the goal is to swim forward as fast as possible.

**Finger Spin.** The agent is a robot with two fingers, and there's a freely spinning object. The goal is to keep the object spinning and balanced on the fingertips. The state includes the positions and velocities of the fingers and the object, and the actions are the forces applied by the fingers.

**Walker Walk.** The agent is a bipedal robot, and the goal is to walk forward as fast as possible. The agent's state includes the angle and angular velocity of the torso, and the joint angles and velocities. The actions are the torques applied to the joints.

**Fish Upright.** The agent is a fish-like robot swimming in a three-dimensional fluid. The goal is to swim forward while maintaining an upright orientation. The agent's state includes the orientation and velocity of the fish, and the actions are the torques and forces applied to move the fish.

**Quadruped Walk.** The agent is a quadrupedal (four-legged) robot. Like the bipedal walker, the goal is to walk forward as fast as possible. The agent's state includes the angle and angular velocity of the torso, and the joint angles and velocities. The actions are the torques applied to the joints.

To ensure consistency in our conclusion, the experiments are conducted under various non-stationarity settings, which include 'within-episode & across-episode', 'within-episode', and 'across-episode' non-stationarities. These settings are respectively denoted as (W+A)-EP, W-EP, and A-EP. Specifically, these non-stationarities can be expressed as

$$s' = f(s,a) + f(s,a) \cdot \alpha_d \left[ c_1^t \cos(c_2^t \cdot t) + c_3^i \sin(c_4^i \cdot i) \right] \tag{D.1}$$

$$s' = f(s,a) + f(s,a) \cdot \alpha_d \left[ c_1^t \cos(c_2^t \cdot t) \right] \tag{D.2}$$

$$s' = f(s,a) + f(s,a) \cdot \alpha_d \left[ c_3^i \sin(c_4^i \cdot i) \right] \tag{D.3}$$

We experimented only under (W+A)-EP when looking at the performance of the algorithm, while in a more detailed ablation study we experimented with all three different settings.

### D.2    SETTINGS OF BASELINES

For VariBAD, we meta-train the models (5000 batch size, 2 epochs for all experiments) and show the learning curves of meta-testing. The tasks parameters for meta-training are uniformly sampled from a Gaussian distribution $\mathcal{N}(0, 1)$.

For all approaches, we use the same backbone algorithm for policy optimization, *i.e.*, PPO with the same hyperparameters, as shown in Table E.2.

### D.3    FULL RESULTS OF PERFORMANCE

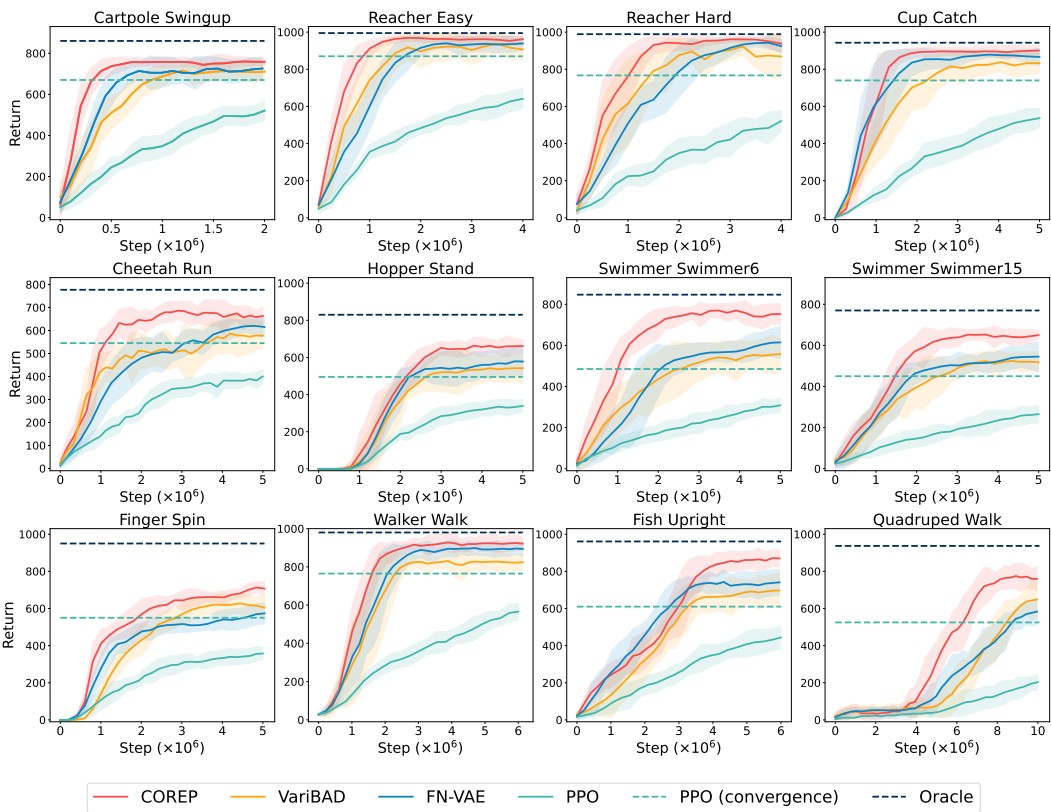

**Figure D.2.** Learning curves of COREP and baselines in different environments. Solid curves indicate the mean of all trials with 5 different seeds. Shaded regions correspond to standard deviation among trials. The dashed lines represent the asymptotic performance of PPO and Oracle.

Figure D.2 shows the full learning curves. We add non-stationary noise as Equation D.1 to all environments. According to the results, COREP consistently performs well in environments of different complexities, proving the effectiveness of the algorithm. Especially in *Hopper Stand, Swimmer Swimmer6, Swimmer Swimmer15, Finger Spin, Fish Upright, and Quadruped Walk*, COREP demonstrates a larger performance gap, highlighting its superiority over baselines.

FN-VAE has the ability to approach our COREP in some simple environments (*Cartpole Swingup, Reacher Easy, Reacher Hard, and Cup Catch*), but still exhibits significant variance, reflecting its instability, especially in more complex environments where it performs even worse than VariBAD (*Finger Spin, Quadruped Walk*). VariBAD shows a large performance gap and variance in all environments, indicating poor stability to non-stationarity. PPO's performance is consistently the worst across all environments due to the lack of any optimization for non-stationarity.

## D.4 FULL RESULTS OF ABLATION STUDY

**Figure D.3.** Final mean returns of 3 different trials on all environments with different components and non-stationarity settings. Returns are normalized to the full version of COREP in each environment.

Figure D.3 shows the performance after removing different components in COREP. All (W+A)-EP, W-EP, and A-EP non-stationary noises (Equation D.1, D.2, and D.3) are separately added to the environments. Each point of Figure D.3 represents the normalized return, which is used to observe the contribution of removed components to the overall algorithm. Specifically,

- w/o VAE is a version without the VAE structure;
- w/o $\mathcal{L}_{\text{guide}}$ removes the guided update mechanism containing TD detection;
- w/o $\mathcal{L}_{\text{sparsity}}$ removes the corresponding loss $\mathcal{L}_{\text{sparsity}}$ in Equation 3.6;
- w/o $\mathcal{L}_{\text{MAG}}$ removes the corresponding loss $\mathcal{L}_{\text{MAG}}$ in Equation 3.6;
- w/o COREP represents the complete removal of all COREP components.

According to the results, it can be seen that the removal of the overall structure of COREP shows a consistent performance degradation, which indicates its important role in tackling non-stationarity. The guided update mechanism and VAE structure also have significant impacts on the algorithm's performance in non-stationary environments. $\mathcal{L}_{\text{sparsity}}$ and $\mathcal{L}_{\text{MAG}}$ also provide some degree of assistance in dealing with non-stationarity.

## D.5 Full Results on Non-stationarity Degrees

**Figure D.4.** Final mean returns of 3 different trials on Cheetah Run and Swimmer Swimmer6 environments with different degrees of non-stationarity. Returns are normalized to the COREP algorithm with standard degree 1.0.

To analyze the impact of varying degrees of non-stationarity in the environment, we change the values of $\alpha_d$ in Equation D.1. As depicted in Figure D.4, the results suggest that the performance of the compared baselines is more affected by the degree of non-stationarity. Conversely, COREP exhibits consistent performance when encountering different degrees of non-stationarity, further affirming our claim that COREP can effectively tackle more complex non-stationarity.

## D.6 STUDY ON USING SINGLE GAT STRUCTURE

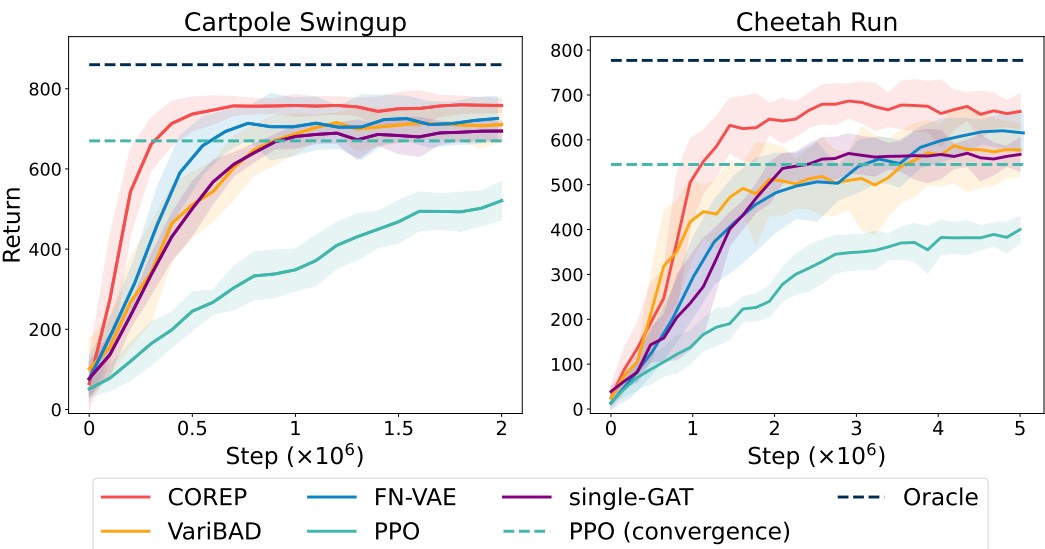

**Figure D.5.** Learning curves of COREP, single-GAT variant, and baselines in different environments. Solid curves indicate the mean of all trials with 5 different seeds. Shaded regions correspond to standard deviation among trials. The dashed lines represent the asymptotic performance of PPO and Oracle. It can be seen that the single-GAT variant exhibits significant performance decreases when facing non-stationarity.

## D.7 STUDY ON TUNNING PARAMETERS OF LOSS TERMS

**Table D.1.** Sensitivity results of parameter $\lambda_{\text{guide}}$. The results indicate that the performance is not sensitive to $\lambda_{\text{guide}}$, even with significant adjustments to its value, COREP still outperforms FN-VAE.

| $\lambda_{\text{guide}}$ | Cartpole Swingup | Reacher Easy | Reacher Hard | Cup Catch | Cheetah Run | Hopper Stand |
|---|---|---|---|---|---|---|
| 0.5 | $732.3 \pm 30.6$ | $947.6 \pm 24.6$ | $938.4 \pm 29.2$ | $868.5 \pm 24.7$ | $634.8 \pm 41.6$ | $651.4 \pm 29.3$ |
| 0.1 (original) | $\mathbf{743.4 \pm 21.2}$ | $\mathbf{964.6 \pm 17.3}$ | $947.2 \pm 23.1$ | $\mathbf{877.5 \pm 19.2}$ | $\mathbf{651.1 \pm 44.3}$ | $645.5 \pm 25.8$ |
| 0.05 | $732.8 \pm 26.6$ | $939.3 \pm 16.4$ | $\mathbf{949.4 \pm 25.7}$ | $870.1 \pm 19.8$ | $642.7 \pm 52.8$ | $\mathbf{656.7 \pm 32.3}$ |
| 0.01 | $722.3 \pm 28.4$ | $945.2 \pm 23.3$ | $934.3 \pm 19.5$ | $864.6 \pm 25.5$ | $629.4 \pm 57.2$ | $641.3 \pm 33.8$ |
| 0.001 | $717.5 \pm 31.9$ | $928.8 \pm 25.8$ | $936.9 \pm 25.7$ | $858.3 \pm 18.6$ | $622.8 \pm 49.2$ | $627.5 \pm 27.5$ |
| FN-VAE | $710.3 \pm 64.5$ | $913.3 \pm 38.7$ | $928.1 \pm 21.9$ | $851.3 \pm 31.6$ | $606.5 \pm 75.3$ | $580.9 \pm 47.3$ |

**Table D.2.** Sensitivity analysis of parameters $\lambda_{\text{VAE}}$, $\lambda_{\text{sparsity}}$, and $\lambda_{\text{MAG}}$. In the experiment, we followed the same setting as in the paper and set these three parameters to the same value for tuning together. The results indicate that the performance is not sensitive to $\lambda_{\text{VAE}}$, $\lambda_{\text{sparsity}}$, and $\lambda_{\text{MAG}}$, even with significant adjustments to its value, COREP still outperforms FN-VAE.

| $\lambda_{\text{VAE}}/\lambda_{\text{sparsity}}/\lambda_{\text{MAG}}$ | Cartpole Swingup | Reacher Easy | Reacher Hard | Cup Catch | Cheetah Run | Hopper Stand |
|---|---|---|---|---|---|---|
| 0.01 | $728.7 \pm 36.1$ | $936.3 \pm 23.7$ | $931.5 \pm 24.4$ | $864.9 \pm 25.8$ | $632.2 \pm 37.5$ | $623.5 \pm 32.1$ |
| 0.005 | $\mathbf{745.6 \pm 34.8}$ | $955.3 \pm 25.3$ | $\mathbf{953.6 \pm 22.1}$ | $872.5 \pm 21.5$ | $644.5 \pm 36.1$ | $638.1 \pm 24.6$ |
| 0.001 (original) | $743.4 \pm 21.2$ | $\mathbf{964.6 \pm 17.3}$ | $947.2 \pm 23.1$ | $\mathbf{877.5 \pm 19.2}$ | $\mathbf{651.1 \pm 44.3}$ | $\mathbf{645.5 \pm 25.8}$ |
| 0.0005 | $718.6 \pm 31.5$ | $946.6 \pm 19.6$ | $945.5 \pm 19.4$ | $856.3 \pm 18.7$ | $621.6 \pm 48.4$ | $641.1 \pm 29.6$ |
| 0.0001 | $721.1 \pm 35.9$ | $949.5 \pm 27.8$ | $940.6 \pm 19.7$ | $859.2 \pm 26.4$ | $618.1 \pm 55.7$ | $619.5 \pm 31.7$ |
| FN-VAE | $710.3 \pm 64.5$ | $913.3 \pm 38.7$ | $928.1 \pm 21.9$ | $851.3 \pm 31.6$ | $606.5 \pm 75.3$ | $580.9 \pm 47.3$ |

### D.8 VISUALIZATION OF LEARNED GRAPH

To better visualize the graph learned by the COREP algorithm, we respectively show the weighted adjacency matrix of core-GAT and general-GAT after 5M steps in the Cheetah Run environment in Figure D.6a and Figure D.6b, and in the Walker Walk environment in Figure D.7 and Figure D.8. The number of graph nodes is a hyperparameter determined by the complexity of the environment. Please refer to Table E.1 for different settings in each environment.

It is important to note that the nodes in the graphs represent abstract components rather than the actual dimensions of the environment, thus direct interpretation of specific graph parts as corresponding to the actual environment is not feasible. However, we can still infer an abstract graph structure. Specifically, in these heatmaps, each value on a grid represents the weight of an edge from a node on the y-axis to a node on the x-axis. A higher value indicates a greater causal influence.

Based on the results, it can be seen that core-GAT focuses more on a few core nodes in its learned graph structure, while general-GAT compensates for some overlooked detailed information by core-GAT. The results align well with our claim made in the paper.

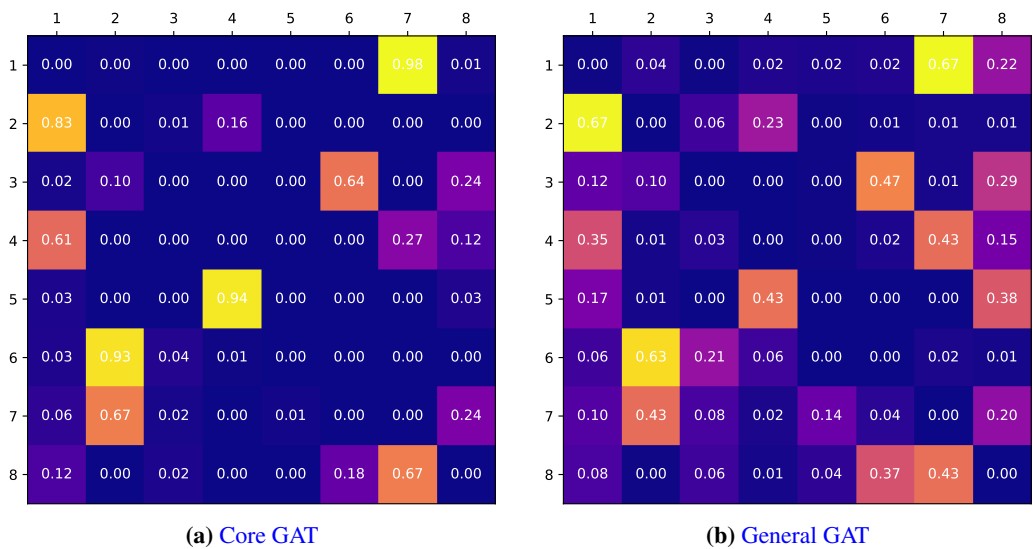

(a) Core GAT      (b) General GAT

**Figure D.6.** Weighted adjacency matrix in Cheetah Run after 5M steps.

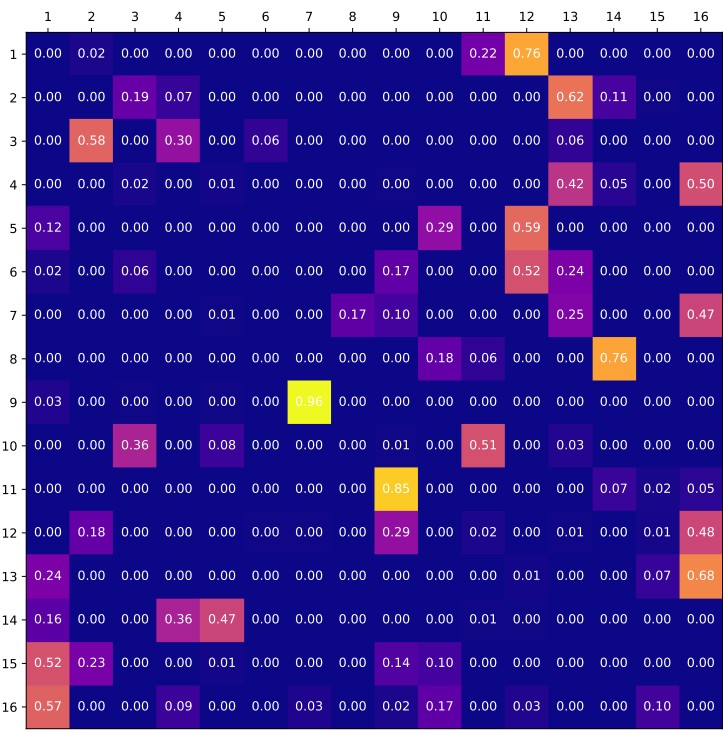

**Figure D.7.** Weighted adjacency matrix of core-GAT in Cheetah Run after 5M steps.

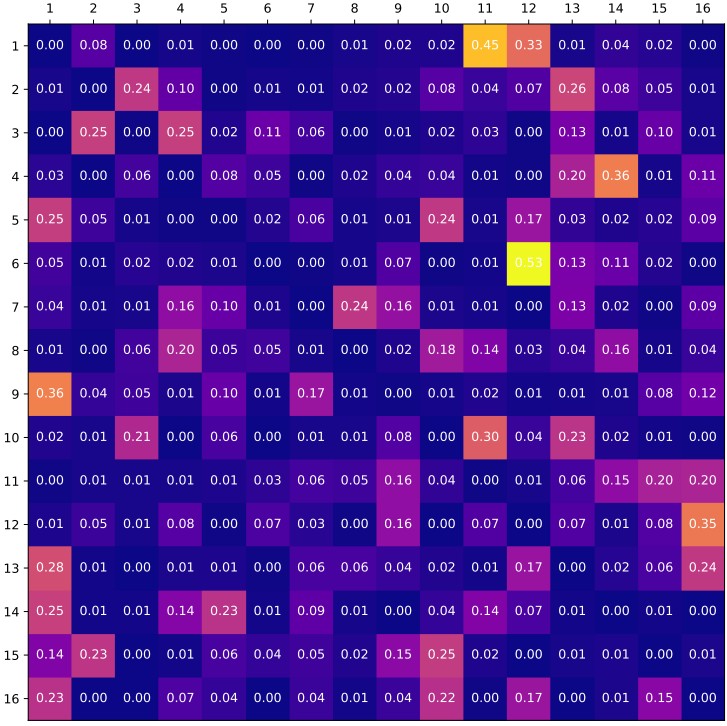

**Figure D.8.** Weighted adjacency matrix of general-GAT in Cheetah Run after 5M steps.

# E IMPLEMENTATION AND TRAINING DETAILS

## E.1 PSEUDO CODE FOR COREP

In Algorithm E.1, we summarize the steps of COREP. For more specific details, please refer to the code provided in our supplementary material.

---

**Algorithm E.1** **C**ausal-**O**rigin **REP**resentation (**COREP**)

---

1: **Init:** env; VAE parameters $\theta, \phi$; policy parameters: $\psi$; replay buffer $\mathcal{B}$; TD buffer $\mathcal{B}_\delta$.
2: **for** $i = 0, 1, \ldots$ **do**
3:      Collect trajectory $\tau_i$ with $\pi_\psi(\boldsymbol{a}|\boldsymbol{s}, \boldsymbol{h})$.
4:      Update replay buffer $\mathcal{B}[i] \leftarrow \tau_i$.
5:      **for** $j = 0, 1, \ldots$ **do**
6:          Sample a batch of episodes $E_j$ from $\mathcal{B}$ and TD errors $\{\delta_k\}$ from $\mathcal{B}_\delta$.
7:          Transform states into $\boldsymbol{X}$ through MLPs.
8:          Compute $\boldsymbol{A_X} = \text{Softmax}\left(\boldsymbol{X}\boldsymbol{X}^{\mathrm{T}} \odot (\boldsymbol{1}_N - \boldsymbol{I}_N)\right)$.
9:          Compute $\delta_\alpha = \left(\sum_{|\mathcal{B}_\delta| - \alpha|\mathcal{B}_\delta| < k < |\mathcal{B}_\delta|} \delta_k\right) / \alpha|\mathcal{B}_\delta|$.
10:          **if** $\delta_\alpha \notin (\mu_\delta - \eta\sigma_\delta, \mu_\delta + \eta\sigma_\delta)$ **then**          ▷ Confidence detection of TD error.
11:             unfreeze weights of core-GAT.
12:          **else**
13:             freeze weights of core-GAT.
14:          **end if**
15:          Get graph representation $\boldsymbol{G}_{\text{core}}, \boldsymbol{G}_{\text{general}}$ from core-GAT and general-GAT.
16:          Compute $\mathcal{L}_{\text{guide}}, \mathcal{L}_{\text{MAG}}, \mathcal{L}_{\text{sparsity}}$ according to Equation 3.4, 3.5.
17:          Input $\boldsymbol{G}(\boldsymbol{s}) = \boldsymbol{G}_{\text{core}} \oplus \boldsymbol{G}_{\text{general}}$ into VAE encoder $q_\phi$ and infer $\mu_{\boldsymbol{h}}, \sigma_{\boldsymbol{h}}$.
18:          Sample $\boldsymbol{h} \sim \mathcal{N}(\mu_{\boldsymbol{h}}, \sigma_{\boldsymbol{h}})$
19:          Decode $\hat{\boldsymbol{s}}$ from $\boldsymbol{h}$ using decoder $p_\theta$, then compute $\mathcal{L}_{\text{VAE}}$ according to Equation B.1.
20:          Do policy optimization for $\pi_\psi(\boldsymbol{a}|\boldsymbol{s}, \boldsymbol{h})$, then compute $\mathcal{L}_{\text{policy}}$ and TD error $\delta$.
21:          Compute $\mathcal{L}_{\text{total}}$ according to Equation 3.6 and use it for gradient-updating $\theta, \phi, \psi$.
22:          Push $\delta$ into TD buffer $\mathcal{B}_\delta$.
23:      **end for**
24: **end for**

---

## E.2 HYPERPARAMETERS

We list the hyperparameters for MLP, GAT, and VAE structures in Table E.1, and the hyperparameters for policy optimization and training in Table E.2.

**Table E.1.** Hyperparameters for the structure of MLP, GAT, and VAE.

| Hyperparameter | Value |
|---|---|
| MLP activation | ReLU |
| MLP hidden dim | 512 |
| MLP learning rate | 1e-3 |
| GAT activation | ELU |
| GAT hidden dim | 32 (Cartpole Swingup, Reacher Easy/Hard, Cup Catch, Cheetah Run) |
| | 64 (Otherwise) |
| GAT node numbers | 4 (Cartpole Swingup, Reacher Easy/Hard, Cup Catch) |
| | 8 (Cheetah Run, Hopper Stand) |
| | 16 (Otherwise) |
| node feature dim | 16 (Cartpole Swingup, Reacher Easy/Hard, Cup Catch, Cheetah Run) |
| | 32 (Otherwise) |
| GAT head numbers | 2 (Quadruped Walk, Fish Upright, Walker Walk, Swimmer Swimmer6/15) |
| | 1 (Otherwise) |
| VAE encoder hidden dim | 128 |
| VAE decoder hidden dim | 64 |
| latent representation dim | 4 (Cartpole Swingup, Reacher Easy/Hard, Cup Catch) |
| | 8 (Cheetah Run, Hopper Stand) |
| | 16 (Otherwise) |

**Table E.2.** Hyperparameters for policy optimization and training.

| Hyperparameter | Value |
|---|---|
| Policy hidden dim | 256 (Swimmer Swimmer6/15, Walker Walk, Fish Upright, Quadruped Walk) |
| | 128 (Otherwise) |
| Policy learning rate | 7e-4 |
| $\lambda_1$ (for $\mathcal{L}_{\text{guide}}$) | 0.1 |
| $\lambda_2$ (for $\mathcal{L}_{\text{MAG}}/\mathcal{L}_{\text{sparsity}}/\mathcal{L}_{\text{VAE}}$) | 1e-3 |
| PPO update epoch | 16 |
| PPO $\gamma$ | 0.97 |
| PPO $\varepsilon$ clip | 0.1 |
| TD buffer size | 2000 |
| Confidence level $\eta$ | 1.96 |

# F    COMPUTE RESOURCE DETAILS

We list the hardware resources used in Table F.1, and list the training time required for a single trial in each environment in Table F.2.

**Table F.1.** Computational resources for our experiments.

| CPU | GPU | RAM |
|---|---|---|
| Intel I9-12900K@3.2GHz (24 Cores) | Nvidia RTX 3090 (24GB) $\times$ 2 | 256GB |

**Table F.2.** Computing time of each single trial in different environments.

| Environment | Training Time |
|---|---|
| Cartpole Swingup | 12 hours |
| Reacher Easy | 16 hours |
| Reacher Hard | 20 hours |
| Cup Catch | 16 hours |
| Cheetah Run | 24 hours |
| Hopper Stand | 28 hours |
| Swimmer Swimmer6 | 28 hours |
| Swimmer Swimmer15 | 36 hours |
| Finger Spin | 28 hours |
| Walker Walk | 28 hours |
| Fish Upright | 30 hours |
| Quadruped Walk | 42 hours |

# G    LICENSES

In our code, we have used the following libraries which are covered by the corresponding licenses:

- Numpy (BSD-3-Clause license)
- PyTorch (BSD-3-Clause license)
- PyTorch Geometric (MIT license)
- DeepMind Control (Apache-2.0 license)
- OpenAI Gym (MIT License)

