# OpenReview forum: "Tackling Non-Stationarity in Reinforcement Learning via Causal-Origin Representation"
_ICLR.cc/2024/Conference — Submitted to ICLR 2024_

### Official Review · Reviewer_gcQg · 2023-10-25

**Soundness:** 3 good
**Presentation:** 2 fair
**Contribution:** 2 fair
**Rating:** 3
**Confidence:** 4

**Summary:**

This work introduces a novel algorithm (COREP) aimed at addressing non-stationarity in reinforcement learning by emphasising causal relationships within states and utilising a "causal-origin" representation. The algorithm can be integrated with existing RL methods, and its effectiveness is supported by theoretical analysis and demonstrated through experiments in diverse non-stationary environments.

**Strengths:**

The main review will take place in this section owing to the flow in which the review was conducted.

### Abstract

- Aren't the first few sentences a bit of a contradiction? You are assuming causal knowledge which is perhaps the strongest prior impractical knowledge there is? But before that you say that modelling the non-stationarity (NS) is impractical because of the strong prior knowledge required.
- Seems like a misnomer to say that we are seeking to 'tackle NS' - surely you are wanting to model it, not treat it like a phenomena that needs to be overcome.

### Introduction

- [second paragraph] It is more correct to say that learning a causal graph from observations in an NS environment is difficult (normal causal discovery is exceptionally difficult in and off itself) because of the inherent distribution shift. This does not necessarily mean that the DAG itself is changing it could be a mechanism drift that is causing the change in dynamics and hence NS. You should explain this.
- I am not entirely sure that you need the last two paragraphs of this section because the information is so dense and specialised, it is okay for the reader to just wait for the main exposition of the method, rather than have you summarise it here in the introduction. I.e. I do not think it adds a lot.

### Methodology

- I do not follow how your method accounts for NS in the mechanism rather than NS in the topology. You are learning a graph topology to model the NS. But the mechanism (the functions, edges, of that DAG) may be generating the concept drift (NS). Consider that in a causal setting, non-stationarity can arise from both changing DAG topology and changing mechanisms within a static DAG topology:
1. **Changing DAG Topology**:
   Non-stationarity can occur when the causal relationships represented by the DAG change over time. For example, in a dynamic causal system, the underlying structure of causal relationships may evolve. This means that the set of variables and their connections, as represented by the DAG, can change. This situation is common in dynamic systems where, for instance, new causal pathways emerge, or existing ones disappear over time. Detecting and modeling these changes in the DAG topology is essential for understanding causality in a dynamic environment.

2. **Changing Mechanisms within a Static DAG Topology**:
   Even within a static DAG topology, non-stationarity can manifest if the mechanisms governing the relationships between variables change over time. This means that the functional forms or conditional dependencies between variables may vary under different conditions or time periods. For example, a medical treatment may have different causal effects on patients of different ages or genders. In this case, the underlying DAG structure remains the same, but the mechanisms, such as the strength of causal connections, can change.

-  Please comment on how your interacts with this setting.
- You assume that there are $K$ distinct stationary environments but $K$ is not upper-bounded. So if $K\rightarrow +\infty$ what happens then?

### Experiments

- [first sentence] Again you are treating NS as adversary rather than something to be modelled. The phrasing seems a bit odd given that you seek a model which can model the NS not remove it.
- Relating back to your abstract, you make the claim that "most existing methods attempt to model changes in the environment explicitly, often requiring impractical prior knowledge" (second sentence). You have not conducted any experiments which actually address that claim but have instead provided (impressive) results which compare your method to other SoTA methods. My main concern with this paper is that I simply do not believe that what you have proposed is in fact _more practical_ than having prior knowledge (as is your claim).

### Related work

- Excellent section. Very comprehensive.

**Weaknesses:**

I am not convinced that this approach is better than the assumption of strong prior knowledge required to model NS in the system. This appears to be far more computationally burdensome.

**Questions:**

See Strengths section.

---

> ### Author Response · Authors · 2023-11-16
>
> We appreciate the reviewer for the insightful and detailed review, which can significantly improve the quality of our paper. Our clarifications to the reviewer's concerns are listed below. We hope the reviewer will be willing to raise the score if the concerns are addressed.
>
> > Aren't the first few sentences a bit of a contradiction? You are assuming causal knowledge which is perhaps the strongest prior impractical knowledge there is? But before that you say that modelling the non-stationarity (NS) is impractical because of the strong prior knowledge required.
>
> It's important to clarify that our approach does not require specific causal knowledge of the environment. Instead, we only assume the existence of causal relationships among environmental variables, without the need for detailed knowledge of their structural interconnections. This is a less demanding assumption compared to the extensive prior knowledge other methods require for modeling non-stationarity (NS). By not necessitating explicit and detailed understanding of the environment, our method stands as a more accessible and feasible approach for tackling NS problems in RL.
>
> > Seems like a misnomer to say that we are seeking to 'tackle NS' - surely you are wanting to model it, not treat it like a phenomena that needs to be overcome.
>
> Our aim is indeed to overcome NS as a phenomenon. The phrase 'tackle NS' is intended to highlight our efforts in addressing the challenges NS poses in RL. Furthermore, our approach to modeling NS aligns with our goal of overcoming this phenomenon, which does not imply a deviation from addressing the problems caused by NS.
>
> > It is more correct to say that learning a causal graph from observations in an NS environment is difficult because of the inherent distribution shift. This does not necessarily mean that the DAG itself is changing it could be a mechanism drift that is causing the change in dynamics and hence NS. You should explain this.
>
> To elucidate this point, let's consider an illustrative example of a simple dynamic system: at time $t_1$, object A influences only object B. However, at a later time $t_2$​, object A exclusively influences object C. In this scenario, the DAG representing the relationships among A, B, and C changes – the edge between A and B disappears, while a new edge between A and C emerges. Furthermore, in more complex dynamics, stochasticity can make these changes less like inherent distribution shifts.
>
> > I am not entirely sure that you need the last two paragraphs of this section because the information is so dense and specialised, it is okay for the reader to just wait for the main exposition of the method, rather than have you summarise it here in the introduction. I.e. I do not think it adds a lot.
>
> We value your suggestion concerning the potential redundancy in the final two paragraphs of our paper's Introduction section. However, we believe that these paragraphs play an important role in setting the stage for our methodology and overall approach. Although the information in these paragraphs is dense and specialized, we have found that some readers appreciate a preliminary overview of the paper's logic and methodology before delving into the details. This initial summary aids these readers in framing their understanding and contextualizing the detailed information presented later. Additionally, these paragraphs enhance the logical flow and coherence of our paper. They serve as a bridge between the general introduction of the problem and the detailed methodology, accommodating the different reading preferences of a diverse audience.
>
> > About how to interact with the setting of changing mechanisms within a static DAG topology.
>
> The setting you described can be considered as a static DAG topology with changing edge weights. As the key idea and the specific method details discussed in section 3.4, we employ a weighted adjacency matrix in GAT to handle this situation. This approach takes into account edge weights during the learning of graph topology. In other words, our reference to changing DAGs encompasses both changes in topology and edge weights.
>
> For better understanding, we have included Appendix D.8 in the revised version, which details the weighted adjacency matrix in the Cheetah Run and Walker Walk environments.

---

> ### Author Response · Authors · 2023-11-16
>
> > You assume that there are K distinct stationary environments but K is not upper-bounded. So if K→+∞ what happens then?
>
> We clarify that the assumption of decomposition into K distinct stationary environments serves to simplify theoretical analysis and provide insights into our algorithm design. It is not intended to establish a framework for practical implementation. In fact, our algorithm doesn't even include the parameter K. Theoretically, as K approaches infinity, this represents a scenario of almost continuous non-stationary changes in the environment. According to our union graph theory, even these nearly continuous changes in the DAG can share a common Mixed Ancestral Graph (MAG) structure. Theoretically, this does not compromise the effectiveness of our designed COREP framework.
>
> In practice, our gradient-updating algorithm requires training only until convergence, even when K is very large. Crucially, our experimental setup, as outlined in equation 4.1, does not necessitate the environment's decomposability. The learning curves shown in our results confirm that COREP effectively converges even under these conditions.
>
> > Relating back to your abstract, you make the claim that "most existing methods attempt to model changes in the environment explicitly, often requiring impractical prior knowledge" (second sentence). You have not conducted any experiments which actually address that claim but have instead provided (impressive) results which compare your method to other SoTA methods. My main concern with this paper is that I simply do not believe that what you have proposed is in fact _more practical_ than having prior knowledge (as is your claim).
>
> We would like to clarify that our paper focuses on addressing the challenges of non-stationarity generally, especially in scenarios lacking specific information about the environment's dynamics. Indeed, existing methods that utilize prior knowledge about environmental changes have some advantages in handling NS. However, we aim to develop a method effective even when such prior knowledge is unavailable or incomplete. This is why our experiments were conducted to compare with baselines developed without prior knowledge. In our experimental setup, all methods, including ours, are subjected to unknown non-stationary changes, providing a fair and relevant comparison.
>
> > I am not convinced that this approach is better than the assumption of strong prior knowledge required to model NS in the system. This appears to be far more computationally burdensome.
>
> Our paper does not assert performance superiority over methods that utilize prior knowledge about the environments. Rather, we concentrate on addressing the impracticality of these methods in broader scenarios, where acquiring detailed prior knowledge is often challenging or unfeasible. Indeed, our approach introduces additional computational demands, but this trade-off is accepted to broaden our capability in addressing NS challenges in RL. In more complicated scenarios, the necessary prior knowledge for NS is often unavailable or unknown, limiting the applicability of methods that heavily rely on such information. By developing a methodology that effectively addresses NS without extensive prior knowledge, we believe our contribution to the field is meaningful.
>
> ----
>
> We hope the above response can address the concerns of the reviewer. If the reviewer still has questions, please feel free to discuss with us! Thanks again to the reviewer for providing us with such helpful suggestions and feedback.

---

> ### Author Response · Authors · 2023-11-21
> **Looking forward to feedback**
>
> Dear Reviewer gcQg,
>
> We again express our gratitude for your valuable and detailed feedback on our work. As the discussion deadline is approaching, we would like to confirm whether our responses above have fully addressed your concerns. We appreciate your comments indicating that a low rating stems from doubts about our method's assumptions and writing. We believe we have clearly addressed these doubts and concerns in our rebuttal. Additionally, we have highlighted our work's main contributions and motivations to help resolve any remaining doubts about our framework.
>
> **If our comments have not fully addressed your concerns, we would greatly appreciate your further feedback.**
>
> Best regards,
>
> Authors

---

> > ### Comment · Reviewer_gcQg · 2023-11-22
> > **Acknowledging**
> >
> > Thanks to the authors for a thorough response. I confirm that I have read their review and have no further questions or comments.

---

### Official Review · Reviewer_RPQF · 2023-10-27

**Soundness:** 3 good
**Presentation:** 3 good
**Contribution:** 1 poor
**Rating:** 3
**Confidence:** 4

**Summary:**

This paper proposes a solution to dealing with a non-stationary environment using causal notions and deep RL.

**Strengths:**

The paper brings together a range of different notions, including causality, deep RL, Causal Structure Discovery, etc.

**Weaknesses:**

The primary issue with the paper is that there are known solutions to this problem that use far simpler approaches than those proposed. The authors are clearly unaware that this is a solved problem. As a consequence, I cannot recommend acceptance without clear comparison with existing solutions, both theoretically and empirically. Theoretically the authors need to justify why such a more computationally complex solution is justified given the existing solutions. Empirically, it is important to know if performance improvements can be obtained wrt state-of-the-art methods.


The paper defines a non-stationary environment " as a nonstationary mixture of various stationary environments".

  --  this is a strong assumption

  --  in control theory there are well-known solutions to systems under such an assumption. For example, multiple model adaptive control is a known solution.

Murray-Smith, R., & Johansen, T. (Eds.). (2020). Multiple model approaches to nonlinear modelling and control. CRC press.

Basically, given a collection of pre-defined stationary environments, if we generate a controller for each of these environments, then we have a guaranteed controller for any mixture of these stationary environments.

Some references about this:

Zhang, W., & Li, Q. (2020). Stable Weighted Multiple Model Adaptive Control of Continuous-Time Plant. In Virtual Equivalent System Approach for Stability Analysis of Model-based Control Systems (pp. 111-127). Singapore: Springer Singapore.

Provan, G., Quinones-Grueiro, M., & Sohége, Y. (2022). Towards Real-Time Robust Adaptive Control for Non-Stationary Environments. IFAC-PapersOnLine, 55(6), 73-78.

Deng, X., Zhang, Y., & Qi, H. (2022). Towards optimal HVAC control in non-stationary building environments combining active change detection and deep reinforcement learning. Building and environment, 211, 108680.

In all of this prior work, no notion of causality is necessary. Hence it is unclear why the causality-based solution proposed is indeed necessary.

Other Issues

Eq. 3.1 is hard to understand
Why not use standard dynamical systems state-space representation? Or probabilistic representation?
if you invent notation, it should be better than what exists. This is not.

Experiments: "we compare COREP with the following baselines: FNVAE (Feng et al., 2022), VariBAD (Zintgraf et al., 2019), and PPO (Schulman et al., 2017)."

---unless you compare COREP with a mixture-based MMAC solution then I cannot see how to understand how well it does. These are only RL-internal methods.

5 RELATED WORK
The authors completely miss known solutions as referenced earlier.

**Questions:**

1. What justifications can be provided for such a complex approach when  mixture-based MMAC already solves the problem? We can use learned models for the model bank in MMAC.

---

> ### Author Response · Authors · 2023-11-16
>
> We thank the reviewer for thoughtful feedbacks and valuable suggestions for our work. To address the reviewer's concerns, we provide detailed responses below. If the concerns can be resolved, we hope the reviewer are willing to raise the score.
>
> > The primary issue with the paper is that there are known solutions to this problem that use far simpler approaches than those proposed. The authors are clearly unaware that this is a solved problem.
>
> We emphasize that the 'non-stationary mixture of various stationary environments' is an idealized theoretical assumption, not a prerequisite for applying the COREP algorithm. As stated in our paper, 'We assume that there are K distinct stationary environments, where K is an unknown number,' which suggests K could be large enough to encompass a near-continuous spectrum of environmental changes. Informed by this insight, COREP aims to learn a stable, shared graph structure, aiding RL algorithms in tackling non-stationarity. Besides, incorporating causality into COREP offers a deeper understanding of environmental dynamics, leading to more robust policy learning, especially in complex and dynamically evolving environments where mixture-based MMAC methods may be less effective.
>
> We reiterate that this assumption serves as a basis for theoretical discussion rather than a constraint on the COREP algorithm. In practice, COREP is effective in environments not strictly adhering to this assumption, as shown in our experiments (eq 4.1). This positions COREP as more versatile than MMAC methods, capable of adapting to a broader range of non-stationary environments. Additionally, in response to the reviewer's suggestion, we have enhanced our revision with a discussion on mixture-based MMAC methods, providing a more comprehensive perspective on existing works in this area.
>
> > Eq. 3.1 is hard to understand Why not use standard dynamical systems state-space representation? Or probabilistic representation? if you invent notation, it should be better than what exists. This is not.
>
> Section 3.1 introduces the key concept of our work: learning the causal-origin graph structure in RL to address non-stationarity, with a focus on causality and stability. To this end, we add binary masks in Eq. 3.1 to denote causal relationships between variables. Specifically, when all binary mask elements are set to 1, indicating causal relationships exist between all variables, Eq. 3.1 then reverts to a standard dynamics function. These masks are crucial for capturing and analyzing causality-driven non-stationarity, and helps causally explain variable relationships and roles in non-stationarity, aligning with our core concept. We argue that this addition does not significantly complicate the understanding. Furthermore, we have strived to ensured that the notation and representation remain accessible and interpretable to readers familiar with standard dynamics in RL.
>
> > unless you compare COREP with a mixture-based MMAC solution then I cannot see how to understand how well it does. These are only RL-internal methods.
>
> We reiterate that our experimental setting is not mixtures of pre-defined stationary environments. As previously discussed, COREP's primary goal is to tackle more complicated non-stationarity in RL. This broader focus on non-stationarity, extending beyond the usual scope of the mentioned mixture-based MMAC solutions, guided our decision to compare COREP with methods such as FN-VAE and VariBAD which are designed for more generalized non-stationary environments. These methods align more closely with the environments and challenges COREP is designed to address.
>
>
> We hope the response resolves the reviewer's concerns. If the reviewer still feels there're something unclear, we're happy to have further discussions! Again we thank the reviewer's time and efforts in making our paper better.

---

> > ### Comment · Reviewer_RPQF · 2023-11-22
> > **Response to rebuttal**
> >
> > Thanks to the author(s) for the response to my review. You may not have fully understood the main point I had. The assumption of  'non-stationary mixture of various stationary environments' leads to a task that is a solved problem, and there are theoretical guarantees for the known solutions. The learning of causal relationships is not necessary to provide such guarantees. The size of K does not matter to this known theory.
> >
> > The authors later state "We reiterate that our experimental setting is not mixtures of pre-defined stationary environments." What precisely is this? The notion "extending beyond the usual scope of the mentioned mixture-based MMAC solutions" is totally unclear. Can you be precise in what you mean? I don't understand " more complicated non-stationarity in RL". There is nothing in the paper or rebuttal to make this precise.

---

> > > ### Author Response · Authors · 2023-11-22
> > >
> > > We thank the reviewer's feedback and hope to precisely address your confusions.
> > >
> > > In the settings used in our work, non-stationary changes are **continuously varying**, meaning that the environment cannot be decomposed into separate mixtures of pre-defined stationary environments. As you mentioned, mixture-based MMAC methods necessitate pre-defined stationary environments, making them unsuitable for the continuously varying non-stationary environment in our work. In such an environment, where **predefined prior knowledge is absent**, mixture-based MMAC methods become unworkable. Therefore, such a setting is **not a solved problem**, and our COREP method is designed to handle these scenarios.
> > >
> > > We again remind the reviewer that our theoretical attempt to decompose the environment into K distinct environments serves merely as **an inspiration or theoretical guide for our algorithm design**, rather than reflecting the actual environment used, where K could theoretically be infinitely large, approximating continuous non-stationarity.
> > >
> > > If you still have any questions, we would greatly appreciate your further feedback.

---

> > > ### Author Response · Authors · 2023-11-22
> > > **supplemental explanation**
> > >
> > > For a more precise explanation, we would like to emphasize that there are **no parameters related to K** in our algorithm. We do not discretize the environment or define hyperparameters related to the environment for the COREP algorithm. Our treatment of the environment is the same as standard RL and **does not require additional knowledge about the environment**.

---

> ### Author Response · Authors · 2023-11-21
> **Looking forward to feedback**
>
> Dear Reviewer RPQF,
>
> We once again express our gratitude for your valuable feedback on our work. As the discussion deadline approaches, we would like to know if our responses above have adequately addressed your concerns. We appreciate your comments regarding the low rating stemming from doubts about the novelty and choice of baselines. We believe we have clearly addressed these doubts and concerns in our rebuttal. Furthermore, we have highlighted the main contributions and motivations of our work, aiming to swiftly resolve any remaining doubts you may have about our framework.
>
> **We would like to know if our comments have fully addressed your concerns, and if not, we are eager to receive further feedback from you.**
>
> Best regards,
>
> Authors

---

### Official Review · Reviewer_TjfN · 2023-10-31

**Soundness:** 1 poor
**Presentation:** 2 fair
**Contribution:** 2 fair
**Rating:** 3
**Confidence:** 4

**Summary:**

This paper addresses nonstationarity in reinforcement learning by positing the presence of a latent variable and utilizing mixture models to model the states. More specifically, the paper introduces an algorithm named COREP, which aims to acquire a stable graph representation for states referred to as the causal origin representation. This formulation can integrate with existing RL algorithms. The experimental results reported in the paper appear to be quite promising.

**Strengths:**

1. This paper introduces a causal interpretation for non-stationary Reinforcement Learning, attributing the nonstationarity to unobserved discrete variables denoted as "e," and it models the states using mixture models. This approach is reasonable.

2. The reported performance surpasses that of other baseline models.

**Weaknesses:**

This paper presents some challenges in terms of clarity and coherence, particularly in explaining how latent state dimensions (denoted as "h") are identified. This identification is crucial for understanding how the MAG is learned over the joint variables {s, h}.

While the authors use bidirected edges in the MAG to capture changes in the marginal distributions of s and h across various environments, it's worth noting that relying solely on bidirected edges may not suffice. It's important to consider that edges with circles might also have latent variable "e," resulting in distribution shifts.

**Questions:**

1. How are the latent state dimensions h identified?

2. Why only consider bidirected edges to indicate the changes?

---

> ### Author Response · Authors · 2023-11-16
>
> We thank the reviewer for the thoughtful review. Below we try to address the concerns of the reviewer. If we are able to resolve the concerns, we hope that the reviewer will be willing to raise the score.
>
> > How are the latent state dimensions h identified?
>
> In Section 3.3, we acknowledge the inherent challenges of theoretically guaranteeing the identifiability of the structure of $\mathcal{M}_{(k)}$ and the latent state dimensions h. This limitation is well-recognized in the study of complex systems like ours. The intricacies of these systems often preclude a straightforward theoretical framework for complete identifiability, especially given the current state of research in this area.
>
> To facilitate intuitive understanding, we have already included a simplified example in Appendix C. This example demonstrates how h might be identified in a simpler scenario. While not a comprehensive solution to identifiability, it serves as a useful illustration of our thought process and methodology.
>
> > Why only consider bidirected edges to indicate the changes?
>
> As discussed in the paper, we adopt the idea of Saeed et al[1] for characterizing causal DAG mixtures, which models them as a family of maximal ancestral graphs (MAGs). This approach represents the observed variables, depicting latent confounders between variable pairs with bidirected edges. For each ancestral graph, a corresponding MAG with the same independence restrictions can be created by adding bidirected edges, as the Theorem 5.1 of [2]. Therefore, our focus will be on graphs with only directed and bidirected edges.
>
> [1] Saeed et al. Causal structure discovery from distributions arising from mixtures of dags. ICML 2020.
>
> [2] Richardson and Spirtes. Ancestral graph markov models. The Annals of Statistics, 2002.
>
> ----
>
> If there are still some remaining concerns, we are happy to have further discussions with the reviewer.

---

> ### Author Response · Authors · 2023-11-21
> **Looking forward to feedback**
>
> Dear Reviewer TjfN,
>
> We again thank you for the positive feedback on the causal interpretation and our performance. As the discussion deadline approaches, we would like to know if our responses have fully addressed your concerns. We appreciate your comments indicating that a low rating stems from doubts about our theoretical analysis. We believe our rebuttal has clearly addressed these doubts and concerns.
>
> **If our comments have not fully addressed your concerns, we would greatly appreciate your further feedback.**
>
> Best regards,
>
> Authors

---

### Official Review · Reviewer_soT9 · 2023-10-31

**Soundness:** 2 fair
**Presentation:** 2 fair
**Contribution:** 2 fair
**Rating:** 8
**Confidence:** 3

**Summary:**

This manuscript introduces an efficient solution to tackle the challenges posed by non-stationary RL, by causality-inspired aspects. This facilitates the necessary adjustments to the policy, enabling it to adapt seamlessly to environmental shifts. The authors employ a graph representation to articulate the data generation process inherent to non-stationary RL systems, introducing a dual graph attention network designed to model both the core-graph and general-graph, which collectively encapsulate the graphical structure of the system. The learning of these graph structures from observed data is central to the identification of causal origins. Empirical evaluations verify the method’s efficacy across a diverse array of non-stationary environments, showcasing its robustness. In essence, this work not only presents a potent solution to non-stationary RL challenges but also holds promising implications for potential applications in related domains, such as transfer RL.

I am also a reviewer of the previous version of this paper. Most of my concerns have been addressed by the authors. However, some of them are still a bit unclear to me. I think the manuscript could benefit from additional clarity, particularly regarding the theoretical underpinnings of causality, and a more comprehensive analysis. Given these considerations, my preliminary assessment aligns with an accept recommendation, contingent upon the aforementioned enhancements.

**Strengths:**

**[About Motivation]** The utilization of causal modeling to understand the environmental aspects of the RL system, and identify the non-stationary origins, is technically sound;

**[About the Algorithm]** The general algorithm design is simple and easy to follow (while certain aspects of the method’s design could be described as empirical, without the grounded theoretical insights);

**[About Writing]** The overall writing is clear and easy to follow.

**[About Experiments]** The experimental design is comprehensive, including a variety of common non-stationary RL environments, as well as diverse factors of change.

**Weaknesses:**

I listed both the weaknesses and questions here.

**[About the theoretical insights of the proposed method]**

From the standpoint of empirical design, I can grasp the concept of amalgamating two graphs to extract the authentic causal graph. However, there remains a lack of clarity regarding the assurance that this specific design is capable of successfully distilling the causal origin of the changes observed. Providing theoretical insights or references that elucidate this process would contribute greatly to the understanding of the methodology. Alternatively, empirical validations, such as demonstrating a correspondence between the graphs generated by the model and the actual causal origins as defined within a simulator, would also serve to reinforce the credibility and effectiveness of the proposed design.

**[About graphs]**

If it is possible to empirically display the learned representation in both graphs? I understand that the environments used in the paper are complicated, and may lack ground truth to find a link between the learned graph representation to the real system parameters. However, the authors could consider doing it on some simulated MDPs (adding non-stationary factors to them) to validate the learned representation.

**Questions:**

I listed both the weaknesses and questions in the above section.

---

> ### Author Response · Authors · 2023-11-16
>
> We thank the reviewer for the positive comment and valuable suggestions for our work. To address the reviewer's concerns, we have conducted additional experiments and revised the paper. The following are detailed responses.
>
> > Providing theoretical insights or references that elucidate this process would contribute greatly to the understanding of the methodology. Alternatively, empirical validations, such as demonstrating a correspondence between the graphs generated by the model and the actual causal origins as defined within a simulator, would also serve to reinforce the credibility and effectiveness of the proposed design.
>
> We have already provided a directly computable, simple example in Appendix C of our paper, which clearly illustrates the process of tracing the causal origin of non-stationarity, highlighting its key aspects. To facilitate a better understanding, we have enhanced Appendix C with more detailed explanations in our revision. We hope this example can aid the reviewer in better understanding this process.
>
> > If it is possible to empirically display the learned representation in both graphs?
>
> We output the weighted adjacency matrices of core-GAT and general-GAT after 5 million steps in the Cheetah Run and Walker Walk environments. For detailed results and explanations, please see Appendix D.8 and Figures D.6, D.7, D.8 in the revised version of our paper. It is important to note that the nodes in the graphs represent abstract components rather than the actual dimensions of the environment, thus direct interpretation of specific graph parts as corresponding to the actual environment is not feasible. However, the results can help us find that the core graph focuses on key elements, and the general graph captures the part potentially overlooked by the core graph, while still maintains the overall structure. These results validate our claim in the paper.
>
> ----
>
> We hope that these explanations and experimental results can address your concerns. If there are still unresolved issues, please feel free to have further discussion with us! Thanks again to the reviewer for the time and effort in improving our paper!

---

### Official Review · Reviewer_7zHg · 2023-11-04

**Soundness:** 2 fair
**Presentation:** 1 poor
**Contribution:** 2 fair
**Rating:** 3
**Confidence:** 3

**Summary:**

The paper addresses non-stationarity in reinforcement learning, highlighting its amplification through causal relationships during state transitions. Instead of directly modeling environmental changes, the authors introduce the Causal-Origin REPresentation (COREP) algorithm. COREP uses a guided updating mechanism to create a stable state representation, making policies more resilient to non-stationarity. Experimental results support COREP's effectiveness over existing methods.

**Strengths:**

1. The paper considers an important problem: addressing non-stationarity in reinforcement learning.

2. The authors provide a good literature review on three different techniques this paper utilized.

**Weaknesses:**

1. This paper integrates three different techniques but the underlying motivation is unclear.

- The choice of GNNs (specifically GAT) for representing causal graphs necessitates a stronger justification, given that causality inherently relies on the structure of directed acyclic graphs.

- Similarly, the adoption of causal-origin representation within the reinforcement learning framework also needs more justifications.

2. One of my biggest concerns lies in the theoretical analysis. As the paper is concerned with the non-stationary RL, the performance guarantees, i.e., regret bound and convergence rate, are not studied at all.

3. Another biggest concern is that the novelty and contribution of this work are not significant. The GAT, causal discovery, and non-stationary RL are well studied in many existing literature. The paper does not provide significantly novel methodological or theoretical techniques in the current stage.

4. The mixing of varying components for policy optimization makes the algorithm difficult to train. I am concerned about the efficiency and complexity of the proposed method, especially compared to the existing baselines (in terms of model complexity, computation, and sample efficiency).

5. The total loss contains multiple tuning parameters (>3), how to efficiently tune these parameters, and what's the sensitivity of the model performance concerning these parameters? The authors only present some preliminary sensitivity analyses on tasks such as cartpole swingup, it is insufficient to demonstrate the robustness of these parameters.

6. The training procedure is not fully disclosed. The authors need to provide specific learning rates, initial values set up, parameter-tuning details, and values for each task.

7. The experimental design to emulate non-stationarity is somewhat naive and may not encapsulate the variety and complexity of non-stationarities encountered in practical settings. The algorithm's efficacy under more typical real-world conditions remains unverified, thus limiting the potential impact of the work.

8. The presentation of this paper needs significant improvement. And I highly recommend the authors do more proofreading. In particular:

- The format of notations is not standard in RL, causal inference, or GNN literature, which puts additional burdens on readers to understand.

- The logical flow of the paper is not clear, and the central claim is vague.

- There exist many typos and errors, like the repeated authors' names (redundant citations) in the section of related works.

**Questions:**

Please consider addressing the weaknesses above.

---

> ### Author Response · Authors · 2023-11-16
>
> We greatly appreciate the reviewer's valuable and constructive review on our work, which significantly improves the quality of our paper. We provide responses to each of your concerns as below. We hope the reviewer can consider raising the score if the concerns are resolved.
>
> > The choice of GNNs (specifically GAT) for representing causal graphs necessitates a stronger justification, given that causality inherently relies on the structure of directed acyclic graphs.
>
> Our paper chooses GAT to represent causal graphs due to its efficient handling of DAGs. This aligns with causality's inherent nature, typically depicted in DAGs. GAT's ability to prioritize neighboring nodes through its masked attention mechanism enables the effective capture of complex causal relationships among state elements, which is crucial for our research's core challenge: tracing the causal origins of non-stationarity. The suitability of GAT for this purpose is further elaborated in our paper, particularly in the section discussing the transformation of observed states into node matrices and the generation of weighted adjacency matrices for core-GAT and general-GAT (reference: Section 3.1).
>
> > Similarly, the adoption of causal-origin representation within the reinforcement learning framework also needs more justifications
>
> We emphasize that the causal-origin representation, a novel design proposed in this paper, is a significant contribution of our research, not an existing method adopted into our work. This design is detailed extensively in Sections 3.1, 3.2, and 3.3, with a thorough theoretical analysis in Appendix A. As outlined in Section 3.1, the causal-origin representation provides RL algorithms with a stable, causally-oriented state representation. This protects RL algorithms from the adverse effects of non-stationarity, enhancing their robustness and effectiveness in complex environments. Consequently, this design significantly enhances the RL algorithms' ability to tackle complex non-stationarity.
>
> > One of my biggest concerns lies in the theoretical analysis. As the paper is concerned with the non-stationary RL, the performance guarantees, i.e., regret bound and convergence rate, are not studied at all.
>
> We agree with the importance of performance guarantees such as regret bounds and convergence rates in RL research. However, we clarify the focus and scope of our theoretical analysis in this context. Our primary goal is to establish a method for providing non-stationarity-informed state representations for RL algorithms. The core of our analysis validates our COREP algorithm's design and efficacy. This focus is crucial, providing theoretical support for COREP's rationality and offering key insights into our algorithm's design.
>
> Additionally, our work, while theoretical in parts, is not purely theoretical. Consistent with much academic research, we substantiate our framework's performance through empirical experimentation. Our results demonstrate that COREP achieves considerable performance improvements across various environments. This approach aligns with field norms, where complex models are often validated through experimental evidence rather than purely theoretical guarantees. While regret bounds and convergence rates are important, they represent a different aspect of theoretical analysis that is not our primary focus. Our emphasis on causal-origin representation and its performance in non-stationary environments is substantiated through rigorous experimentation, which we believe provides compelling evidence of our algorithm's effectiveness.
>
> > Another biggest concern is that the novelty and contribution of this work are not significant. The GAT, causal discovery, and non-stationary RL are well studied in many existing literature. The paper does not provide significantly novel methodological or theoretical techniques in the current stage.
>
> We reiterate the key idea of our research, highlighting its originality and significance: the design of the causal-origin representation.
> 1. **Innovation in Causal-Origin Representation:** Contrary to the implication of merely adopting existing methods, we emphasize that the causal-origin representation is an original concept, conceived and developed within this work. Far from being a minor addition, it forms the core of our theoretical analysis and methodological design, representing a significant advancement in the field of non-stationary RL.
> 2. **Contribution to Non-Stationary RL:** The introduction of causal-origin representation addresses a critical gap in non-stationary RL. It provides a novel approach for examining and responding to the complexities of non-stationary environments, diverging from conventional methods focused on post-change detection or latent variable modeling of change factors. This innovation not only enhances current understanding but also paves the way for future advancements in handling dynamic environments in RL.

---

> ### Author Response · Authors · 2023-11-16
>
> > The mixing of varying components for policy optimization makes the algorithm difficult to train. I am concerned about the efficiency and complexity of the proposed method, especially compared to the existing baselines (in terms of model complexity, computation, and sample efficiency).
>
> We would like to address these concerns by highlighting key aspects of our approach detailed in our paper:
>
> 1. **Integration of VAE for Enhanced Learning Efficiency:** In Section 3.4, we discuss integrating a Variational AutoEncoder (VAE) into the COREP framework to improve learning efficiency. The VAE helps manage the complexities of the causal-origin representation, enhancing learning efficiency.
> 2. **Empirical Evidence of Sample Efficiency:** Figure 4.1 demonstrates COREP's sample efficiency advantage over baselines. These results confirm that COREP maintains, and often surpasses, the sample efficiency of existing methods, despite integrating multiple components.
> 3. **Computational Resource Utilization:** Appendix F provides detailed information about computational resources and time. The results demonstrate COREP's normal computational level in environments like the DeepMind Control Suite, evidencing its computational feasibility in standard RL settings.
> 4. **Limitations in High-Dimensional Environments:** As Section 6 outlines, we recognize COREP's potential complexity issues in high-dimensional environments. Addressing this in future work, we aim to optimize COREP's scalability and performance in such settings. However, this limitation is beyond the current work's scope and does not detract from COREP's contributions and effectiveness in tested environments.
>
> > The total loss contains multiple tuning parameters (>3), how to efficiently tune these parameters, and what's the sensitivity of the model performance concerning these parameters? The authors only present some preliminary sensitivity analyses on tasks such as cartpole swingup, it is insufficient to demonstrate the robustness of these parameters.
>
> - Contrary to the concern raised, our total loss function (equation 3.6) involves only two tuning parameters, $\lambda_1,\lambda_2$. We were aware of potential complexity arising from multiple modules, thus we intentionally aggregated different loss components by magnitude. This led to retain only two coefficients for tuning, simplifying parameter tuning.
> - We further demonstrated COREP's low parameter dependency in the sensitivity analysis in Appendix D.7. Despite significant adjustments to these parameters, COREP's performance still surpasses the SOTA baseline FN-VAE.
> - Following the reviewer's suggestion, our sensitivity analysis now includes a broader range of environments. The revised analysis, highlighted in blue in Appendix D.7, includes this expanded range and validates our claim with consistent results.
>
> > The training procedure is not fully disclosed. The authors need to provide specific learning rates, initial values set up, parameter-tuning details, and values for each task.
>
> Information on learning rates, initial values, and parameter-tuning details for each task is already detailed in Table E.1 and Table E.2 of our paper. These tables provide a comprehensive outline of the hyperparameters used in our experiments, including those for the structure of MLP, GAT, and VAE, as well as the policy optimization and training hyperparameters. We believe that these tables comprehensively address your concerns regarding the specific parameters used in our experiments.
>
> > The experimental design to emulate non-stationarity is somewhat naive and may not encapsulate the variety and complexity of non-stationarities encountered in practical settings. The algorithm's efficacy under more typical real-world conditions remains unverified, thus limiting the potential impact of the work.
>
> - In Section 4, under Experiment Settings, we mention our introduction of non-stationarity through periodic noise in the dynamics, aligning with the SOTA method FN-VAE. This representation of non-stationarity is common in similar field research [1, 2]. We adopted this setting to ensure a fair and comparable performance assessment across methodologies.
> - While this approach may not capture all complexities of real-world non-stationarity, it provides a suitable benchmark for exploratory research like ours. The primary goal of our research is to validate new methodological designs in widely-used academic environments. We are confident that our chosen settings effectively demonstrate COREP's capability to surpass SOTA methods. We agree that testing our algorithm in more complex, real-world settings is a valuable direction for future research.
>
> [1] Feng et al. Factored adaptation for non-stationary reinforcement learning. NeurIPS 2022.
>
> [2] Xie et al. Deep reinforcement learning amidst lifelong nonstationarity. ICML 2021.

---

> ### Author Response · Authors · 2023-11-16
>
> > The format of notations is not standard in RL, causal inference, or GNN literature, which puts additional burdens on readers to understand.
>
> - The main modification we made to the notation is simply adding binary masks to the variables of RL dynamics in Equation 3.1, which are used to define causal relationships between elements. This modification is necessary for the subsequent introduction and analysis of our method. Particularly, when all elements have a mask value of 1, Equation 3.1 degenerates into a standard RL dynamics function.
> - In our discussion of the union graph in Sections 3.2 and 3.3, we adhere to standard graph theory notations.
> - In Section 3.4, our description of GAT details strictly follows the notation system prevalent in GNN literature.
>
> We believe that our notation format is consistent with respective fields and should not impose additional burdens on the readers.
>
> > The logical flow of the paper is not clear, and the central claim is vague.
>
> Our paper's logic begins in the introduction, where we discuss the motivation for using causal information in representations to help RL algorithms address non-stationarity. Following this, we define non-stationarity from a causal perspective and conduct a theoretical analysis. This leads to the innovative design of the union graph, proven to encapsulate the defined non-stationarity information. Building on these insights, we develop the COREP algorithm, which learns the union graph structure and ensures inclusion of non-stationarity information in the graph representation. This design equips RL algorithms with awareness of environmental non-stationarities, effectively addressing the core problem. We believe this progression from problem identification to methodological insights and design is logically coherent.
>
> > There exist many typos and errors, like the repeated authors' names (redundant citations) in the section of related works.
>
> Regarding the redundant names in the citation format in related works, we appreciate the reviewer for pointing it out, and we have made adjustments in the revision. We also further thoroughly examine the paper to make it accurate and rigorous.
>
> ----
>
> We hope our response can resolve the concerns. If anything remains unclear, please let us know!

---

> ### Author Response · Authors · 2023-11-21
> **Looking forward to feedback**
>
> Dear Reviewer 7zHg,
>
> We are grateful for your recognition of the research background and literature review in our work. As the discussion deadline approaches, we would like to ensure that our responses above have adequately addressed your concerns. We appreciate your comments regarding doubts about motivation and novelty, which we believe have been clearly addressed in our rebuttal. Additionally, we have highlighted our work's main contributions and motivations to further clarify any doubts regarding our framework.
>
> **We would like to know if our comments have fully addressed your concerns, and if not, we are eager to receive further feedback from you!**
>
> Best regards,
>
> Authors

---

### Author Response · Authors · 2023-11-16

We thank all the reviewers for the efforts in reviewing our paper and providing insightful suggestions! This has been of great help in improving our paper. In response to the concerns raised by the reviewers, we have provided detailed replies in the responses below. We have also conducted additional experiments and made revisions to the paper based on the reviewers' suggestions. The newly added content is highlighted in blue text within the revised version. Feel free to check the updated PDF paper. If there are still questions, please let us know. We are looking forward to further discussion!

---

### Meta-Review · Area_Chair_dDpn · 2023-12-06

**Metareview:**

The paper proposes Causal-Origin REPresentation (COREP) algorithm, which employs a guided updating mechanism to learn a stable graph representation, and hence enhance robustness to non-stationarity.

pros:
+ The paper addresses the important problem of non-stationarity in reinforcement learning.

cons:
+ lack of clarity on the motivation behind integrating different techniques along with theoretical underpinnings
+ lack of sufficient empirical display of learned representation in graphs
+ lack of sufficient baseline comparisons in empirical studies

**Justification For Why Not Higher Score:**

lack of clarity in motivation and justification; lack of sufficient empirical comparison with baselines

**Justification For Why Not Lower Score:**

NA

---

### Decision · Program_Chairs · 2024-01-16

Reject